# Fault Diagnostics and Tolerance Analysis of a Microgrid System Using Hamilton–Jacobi–Isaacs Equation and Game Theoretic Estimations in Sliding Mode Observers

**DOI:** 10.3390/s22041597

**Published:** 2022-02-18

**Authors:** Ebrahim Shahzad, Adnan Umar Khan, Muhammad Iqbal, Fahad Albalawi, Muhammad Attique Khan, Ahmad Saeed, Sherif S. M. Ghoneim

**Affiliations:** 1Department of Electrical Engineering, FET, International Islamic University, Islamabad 44000, Pakistan; adnan.umar@iiu.edu.pk (A.U.K.); ahmad.phdee85@iiu.edu.pk (A.S.); 2Research and Innovation Centre of Excellence (KIOS CoE), University of Cyprus, Nicosia 1678, Cyprus; muhammad.iqbal@iiu.edu.pk; 3Electrical Engineering Department, College of Engineering, Taif University, P.O. Box 11099, Taif 21944, Saudi Arabia; f.albiloi@tu.edu.sa (F.A.); s.ghoneim@tu.edu.sa (S.S.M.G.); 4Department of Computer Science, HITEC University, Taxila 47040, Pakistan; attique.khan@ieee.org

**Keywords:** microgrids, fault-tolerant control (FTC), current/potential transformer (C.T/P.T), sliding mode observers (SMO), H∞ and H- parameters, Hamilton–Jacobi–Isaacs/Bellman-Equation (HJIE or HJBE), Lyapunov stability, fault diagnosis/estimation, game theory

## Abstract

This paper focuses on robustness and sensitivity analysis for sensor fault diagnosis of a voltage source converter based microgrid model. It uses robust control parameters such as minimum sensitivity parameter (H−), maximum robustness parameter (H∞), and compromised both (H−/H∞), being incorporated in the sliding mode observer theory using the game theoretic saddle point estimation achieved through convex optimization of constrained LMIs. The approach used works in a way that the mentioned robust control parameters are embedded in Hamilton–Jacobi–Isaacs-Equation (HJIE) and are also used to determine the inequality version of HJIE, which is, in terms of the Lyapunov function, faults/disturbances and augmented state/output estimation error as its variables. The stability analysis is also presented by negative definiteness of the same inequality version of HJIE, and additionally, it also gives linear matrix inequalities (LMIs), which are optimized using iterative convex optimization algorithms to give optimal sliding mode observer gains enhanced with robustness to maximal preset values of disturbances and sensitivity to minimal preset values of faults. The enhanced sliding mode observer is used to estimate states, faults, and disturbances using sliding mode observer theory. The optimality of sliding mode observer gains for sensitivity of the observer to minimal faults and robustness to maximal disturbance is a game theoretic saddle point estimation achieved through convex optimization of LMIs. The paper includes results for state estimation errors, faults’ estimation/reconstruction, fault estimation errors, and fault-tolerant-control performance for current and potential transformer faults. The considered faults and disturbances in current and potential transformers are sinusoidal nature composite of magnitude/phase/harmonics at the same time.

## 1. Introduction

Distributed electric generators forming diverse microgrid structures are required to function robustly against faults and disturbances for the reliable working of micro power plants and un-interrupted power delivery to end users. This work focuses on a diagnosis process that is robust to disturbances and sensitive to faults using sliding mode observers enhanced with robust control parameters such as (H−), (H∞) and (H−/H∞) for a better fault reconstruction. The reconstructed faults and unknown inputs/ disturbances can be used for correction of faulty sensor current/potential transformers (C.T/P.T) outputs to enable the control system to act in a fault tolerant way. The work incorporates application of the Hamilton–Jacobi–Isaacs-Equation (HJIE) along with cost functionals of robust control parameters and game theoretic estimations to enhance the conventional sliding mode observer theory.

Hou [1] uses H−/H∞ robust fault detection technique, where the H− index enhances the sensitivity of residual w.r.t fault and H∞ deals with the robustness of disturbances effect on the residual. The observer design is an optimization problem where the necessary conditions for fault detection are embodied in terms of linear matrix inequalities (LMIs), which are solved by an iterative algorithm. Hammouri [2] provides sufficient conditions for fault detection, provided for non-linear affine models along with circumstances in which observers with high gains can be used for residual generation for uniformly observable systems. Edelmayer [3] uses H∞ constraint through a filter transformed into an LMI based convex feasibility optimization problem, to suppress the effect of disturbances and unknown inputs for robust detection of faults and modes of failure in linear time variant (LTV) systems. Zhang [4] uses a detection filter with a bunch of estimators with different thresholds for fault isolation in a dynamic non-linear system with uncertainty. The paper also works on determination of different adaptive thresholds, fault isolation conditions, and analytical results for time required for fault isolation. Liu [5] works on using H− and H∞ norms for worst case fault sensitivity and robustness to disturbances, respectively, represented in the forms of LMIs as bounded real lemma, along with additionally performing the analysis problem for finite frequency range using weighing filters, which is useful in strictly proper systems. Maciejowski [6] gives a detailed non-linear reference model for a crashed aircraft being controlled by model predictive control (MPC) controller and pilot controls being modeled by another MPC controller at low sampling rate to give such a fault detection and isolation (FDI) mechanism, which could have avoided the plane crash in 1862. Yan [7] uses a sliding mode observer based technique for a non-linear air craft system where the uncertainty being the function of the state variables has a non-linear bound. Liu [8] works on making the outputs least sensitive to inputs through a H -index in terms of LMIs added input observability with new conditions necessary and sufficient for fault detection with worst case sensitivity of faults. Edwards and Tan [9,10] use sliding mode observers for fault detection and isolation (FDI) of uncertain linear systems, by output error injection and fault correction through fault detection and estimation while maintaining closed loop performance. Yan [11] addresses fault estimation for bounded specific classed perturbed non-linear systems using the output error injection approach in sliding mode observers (SMOs), where fault can be estimated online to any accuracy and observer parameters are determined by LMIs, and tested for robotic arm. Wang [12] works on H∞H∞ and H−H− index based optimization using LMIs to enhance the detection of faults while attenuating disturbances and uncertainties along with working on the threshold required for fault detection. Wang ([13]) uses H− index for sensitivity to worst case fault detection while minimizing the effect of worst case disturbance using H∞ norm using H−/H∞ observer based fault detection. The conditions necessary and sufficient for fault detection filters are solved with iterative LMI algorithms and results are determined for both the finite and infinite frequency ranges. Wang [14] works on H− and H∞ criteria for fault detection in linear systems, where fault detection problem is unconstrained using a pole assignment approach and observer gains are determined by a gradient optimization approach. The paper considers sensitivity of fault for a finite frequency range. The problem is tested for air craft vertical landing and take-off faults. Pertew [15] designs a dynamic observer for Lipchitz non-linear systems, which uses an objective function through LMIs to converge the residual vector to the fault vector to achieve detection and estimation at the same time. The problem uses convex optimization of objective function by using the suitable weights to minimize the effect of fault on estimation error instead of a conventional constant gain structure. Ding [16] works on fault detection observer scheme where the system is completely decoupled from unknown inputs with a major focus on minimal order fault detection filters along with associated required algorithms for the process. Zhang [17] works on an observer based fault detection with residual norm based minimization on false alarm rate (FAR) for given false detection rate (FDR) using a well-established factorization technique. Khan [18] addresses mainly threshold computation for faults detection, while formulating a problem as an optimization problem using LMIs for continuous time non-linear systems. Alwi [19] uses SMO for estimation of sensor faults, which works even beyond the limitation in terms of the requirement of stability of an open loop system while using other unknown input linear SMOs. Aliyu [20] uses $H_{2}/H\infty$ mixed and finite dimensional filtering while conditions are in terms of coupled discrete Hamilton–Jacobi–Isaacs equations. The problem addresses mainly discrete time nonlinear affine systems while considering both finite and infinite horizon problems. Slim and Dhahri [21] uses Lyapunov stability LMIs based determination of gains of SMO by imposing minimization of H -Infinity criteria (ratio of residual to disturbance), i.e., to minimize the effect of disturbance on reconstruction of fault, which not only improved fault estimation but also validated the work for unmatched uncertainties and faults. Li [22] studies the tradeoff for between sensitivity to faults and robustness to disturbances using H−/H∞ and H∞/H∞ optimization problems using iterative linear matrix inequalities (ILMIs) based on a factorization technique, for linear time invariant (LTI) systems in state space form. Shi [23] uses H-Infinity based fault tolerant control (FTC) for sensor and actuator faults in a wind energy system considering variable wind speed problems using the concepts of Stochastic Affine models and linear quadratic regulator (LQR) state feedback control. Raza [24] considers a switched asynchronous system with disturbances and noises for designing fault detection (FD) filters with mixed H−/H∞ sensitivity and robustness criteria, along with piece wise Lyapunov function stability. Conditions in terms of LMIs and average dwell time were also investigated. The schemes were tested on buck-boost converters for matched and unmatched durations of switched systems. Ahmad [25] uses a robust fault detection filters (FDF) design with H∞ criteria for sensitivity to faults while completely isolating the system from unknown inputs using LMIs for LTI systems. Ning [26] addresses a stochastic system with limited network resources and used filter mechanism for fault detection. The system operates on an event-triggered mechanism instead of a lot of data coming from sensors to filter, which makes the residual generator more sensitive to faults along with disturbance attenuation compared to conventional FD methods along with saving network resources. These are considered in detail [27,28,29,30] in order to learn the approaches. The book of Yuri and Edwards on sliding mode control and observers is also considered in detail [31].

The major contribution of the paper is to incorporate an estimation of game theoretic saddle points similar in nature to robust control parameters (H∞,H−), using their respective cost functionals and inequality version of HJIE in SMO theory, through Lyapunov theory and convex optimization of consequent LMIs, which also ensures stability. The concept is applied on VSC-based microgrid model.In comparison to earlier works, they have used an approach of using robust control parameters and game theory for ordinary Luenberger Observers, whereas this study has incorporated the mentioned approach in sliding mode observers having a Luenberger gain for output estimation error term along with another switch term gain, which ensures more robustness along with suitability for switching electronic systems. Moreover, this work has incorporated game theoretic saddle points through convex optimization of LMIs, unlike earlier works;The approach works mainly for sensor faults of microgrid and particularly for multiple faults occurring simultaneously in their sensor current and potential transformers. The approach works very well for many other types of faults such as square pulse (intermittent), ramp faults (i.e., incipient nature), constant faults, etc., but the results are not included in the paper;The inequality version of HJIE (i.e., Hamiltonian as the inequality constraint), which ensures stability in terms of Lyapunov function (with its negative definiteness) along with giving constrained LMIs. The stability analysis through a cost functional constrained inequality version of HJIE and hence determination of corresponding LMIs are convex optimized to determine the respective SMO gain. The gains are determined for (H−), (H∞) and then compromised of both (H−/H∞) constraints, which are included as theorems as a major contribution of this work;The fault/disturbance estimation by SMO theory, is used for correction of faulty sensor outputs to be supplied to the $P_{r}-Q_{c}$ control block, hence acting additionally as a fault-tolerant control. This part uses the work of the base paper of the authors of this study, referred to in [32].

The block diagram of proposed scheme is presented in Figure 1. It comprises a block containing the Microgrid System Model, along with the PLL Block, injection of faults/disturbances in sensor outputs, and stable filtering, while also providing augmented system states and stable filtered outputs. The ’FD Block’ is state/output estimation SMO, while the FE block performs state/output error estimation SMO. The block tagged ’Reduced Order Fault Estimation’ performs several calculations along along with computation of reduced order state error, the block tagged ’Estimated Sensor Fault Computation’ performs computation of estimated faults/disturbances based on values received from the three above-mentioned SMO blocks. The ’LMI Optimization’ block in the center performs computation of m-files based H−, H∞, and H−/H∞ based SMO gains using the convex optimization of LMIs. The detailed simulink based diagram is presented in Section 5.

Section 1 in this work comprises the introduction, literature review, and the approach of the study. Section 2 gives a brief explanation of the fault model with a brief description of the current/potential transformers, microgrid system model, and stable filtering. Section 3 gives preliminaries of the SMO theory for detection and estimation of faults and unknown inputs (disturbances). Section 4 comprises the proposed work with stability analysis and determination of robust to disturbance and fault sensitive SMO gains. Section 6 presents the results and discussions. Section 7 presents the references whereas the Appendix A includes the proof for the general Schur complement, Hamilton–Jacobi–Bellman-Equation, and Matlab-based Optimization Algorithms, supporting lemmas including proofs of inequality version of HJIE, definitions of Hamiltionian, and an explanation of the Riccati equation and dq/Park/Clark Transformations.

## 2. System Modeling

### 2.1. Mathematical Model of Current/Potential Transformer Faults

The study and its scope is general for many types of sensors and their respective connectivity configurations. However, here the current/voltage transformers of a microgrid are considered as being faulty and perturbed. It considers faults/disturbances occurring in C.T/P.T, which arise from saturation, causing increased core magnetization current, and resulting in a reduced secondary current required for switches and relays. The C.T saturation is caused by a DC offset of fault current on the primary side, prior to the fault remnant flux and cumulative impedance burden of the secondary side, which results from relay coils and even wires. The C.T sensor is like a ring on the wire, which measures the current through magnetic flux and flux density, and hence acts as a current sensor, whereas the P.T is a 1:1 transformer that is used to measure the voltage. Both the current/voltage sensors are installed on an inductor-capacitor-inductor (LCL) filter to measure the input/output currents and voltages. The commonly used technical terms are: Hysteresis, Saturation, and Eddy Currents, etc. [33,34,35]. The inaccurate C.T/P.T output will result in inaccurate space vector pulse width modulation (SVPWM) signals and consequently the faulty voltage source converter (VSC) currents/voltages, which will then not be able to track the desired active and reactive powers. The faults occurring in C.T/P.T are of composite nature, inclusive of magnitude, phase, and harmonics where normally previous works have considered only magnitude faults for the sake of simplicity [36]. The mathematical model of faults and perturbations is serving the existence of the actual C.T/P.T faults comprising of magnitude, phase and harmonics existing at the same time. So, the fault is considered mathematically to be considering magnitude and phase and an additive disturbance being a sinusoidal with different frequency serves as additive harmonics. In this way, the complete model of faults are being considered in the system, which are added using a Simulink block to be added in the faultless VSI current/voltage signals. The fault and disturbance mathematical model is defined as:(1)$$f\left(t\right)+\xi \left(t\right)=f_{o}*sin\left(\omega_{1}t+\phi_{1}\right)+\xi_{o}*sin\left(\omega_{2}t+\phi_{2}\right)$$

### 2.2. Mathematical Model of the Microgrid System

A small signal model of MG is used in this work to determine the workability and efficiency of a fault-tolerant control scheme. The small signal model of MG is claimed to be properly simulated, experimentally verified, and able to be used as a block in large grid networks.

The mathematical model of a voltage source inverter based microgrid basically comprises of non-linear equations, whose linearized version at the operating point is used in this work. The complete microgrid model, as proposed by [37] comprises of inverter equations, LCL filter, droop control equations, PLL, current controller, voltage controller, SVPWM, and real and reactive power calculation, which all together form a non-linear model, and needs to be linearized at the operating point. The LCL filters are placed just after the VSI, as passive filters to manage the current/voltage spikes. The modeling of stray/line inductances, capacitances, and resistances are also considered with LCL filters, i.e., the resistors $r_{c}$ and $r_{f}$ are the parasitic resistances of the inductors, a damping resistor $R_{d}$ is connected in series with the filter capacitor, however, the capacitor’s ESR is not considered, as it can be lumped into $R_{d}$. The current and potential transformers are mounted on LCL filter to read the instantaneous current/voltage readings, which are used to calculate the instantaneous values of real and reactive $\left(P_{r},Q_{c}\right)$ powers. The inverter model is considered an average model in his work, i.e., without any major inaccuracies, we can assume that the commanded voltage appears at the input of the filter inductor i.e., $V_{idq}*=V_{idq}$. This approach neglects only the losses in the IGBT and diodes. The LCL part of the whole system mathematically modeled with KCL/KVL along with average VSI model, is considered as a required part of the complete mathematical model to design the SM observer because it is placed right after VSI, and C.T/P.T are also mounted on them to read the instantaneous values of input/output currents/voltages. Therefore, if there is no fault in the sensors (C.T/P.T) of the system, the model and the actual system states/outputs are compromised and are showing no difference. The Simulink based model is shown in Figure 2.

**Remark** **1.**
*The considered required part of the MG system is a continuous time LTI system, which follows the separable principle, i.e., the controller and observer can work in combination with each other, i.e., the unknown system input/output states required for the control action can be determined using the observer. The pair $\left(A_{s},C_{s}\right)$ are observable, $C_{s}$ is full rank output matrix, the matrices $\left(A_{s},C_{s},E_{s}\right)$ have no invariant zeros, which is satisfied if outputs are more than inputs, i.e., (p>m). For the considered microgrid case p = 4 and m = 2. The fault and disturbance distribution matrices $E_{s}=D_{s}$ are simply the identity matrices in the considered microgrid application.*


**Definition** **1**(Separation Principle). *For the deterministic linear systems, if an observer and a stable state feedback are designed for a linear time-invariant system, then the combined observer and feedback are stable.*

Therefore, the linearized mathematical model of considered MG from [37] is given as under.
(2)$$\left[\begin{array}{c} {\dot{i}}_{i}\\ {\dot{i}}_{o}\\ {\dot{v}}_{o} \end{array}\right]=\left[\begin{array}{ccc} -r_{f}/L_{f} & 0 & -1/L_{f}\\ 0 & r_{c}/L_{c} & 1/L_{c}\\ 1/C_{f}-r_{f}R_{d}/L_{f} & -(1/C_{f}-r_{c}R_{d}/L_{c}) & -(R_{d}/L_{f}+R_{d}/L_{c}) \end{array}\right]\left[\begin{array}{c} i_{i}\\ i_{o}\\ v_{o} \end{array}\right]+\left[\begin{array}{c} 1/L_{f}\\ 0\\ 1/L_{f} \end{array}\right]v_{i}+\left[\begin{array}{c} 0\\ -1/L_{c}\\ R_{d}/L_{c} \end{array}\right]v_{g}$$
where $v_{i}$ is the input (inverter) voltage, $v_{g}$ is the grid voltage, $L_{c}$, $L_{f}$, $C_{f}$ are the LCL filter inductances and capacitances, respectively.

The abc-dq0 transformation is defined as a combination of Clark and Park transformation to convert a three phase voltage/current into effectively a two phase form. Some explanation of the said transformations are given in Appendix A.

For any signal *s*(*t*)
(3)$$\left[\begin{array}{c} s_{d}\\ s_{q}\\ s_{0} \end{array}\right]≜\sqrt{2/3}\left[\begin{array}{ccc} cos(\theta ) & cos(\theta -\frac{2\pi }{3}) & cos(\theta +\frac{2\pi }{3})\\ -sin(\theta ) & -sin(\theta -\frac{2\pi }{3}) & -sin(\theta +\frac{2\pi }{3})\\ \sqrt{2}/2 & \sqrt{2}/2 & \sqrt{2}/2 \end{array}\right]\left[\begin{array}{c} s_{a}\\ s_{b}\\ s_{c} \end{array}\right]$$

A PLL is required to measure the actual frequency of the system. A dq based PLL was chosen, where the PLL input is the d-axis component of the voltage measured across the filter capacitor. Therefore, the phase is locked, such that $V_{od}=0$. The grid side voltage angle is measured by PLL in our problem, and then according to convention that angle is used for all abc-dq and vice versa transformations, which is needed in the system.

The system model in Equation (Equation 2) in *dq* transformed form is
$$A_{s}=\left[\begin{array}{cccccc} \frac{-r_{f}}{L_{f}} & W_{pLL} & 0 & 0 & \frac{-1}{L_{f}} & 0\\ w_{pLL} & \frac{r_{f}}{L_{f}} & 0 & 0 & 0 & \frac{-1}{L_{f}}\\ 0 & 0 & \frac{-r_{c}}{L_{c}} & w_{pLL} & \frac{1}{L_{c}} & 0\\ 0 & 0 & -w_{pLL} & \frac{-r_{c}}{L_{c}} & 0 & \frac{1}{L_{c}}\\ \frac{1}{C_{f}}-\frac{r_{f}R_{d}}{L_{f}} & w_{pLL}R_{d} & \frac{-1}{C_{f}}+\frac{r_{c}R_{d}}{L_{c}} & -w_{pLL}R_{d} & -(w_{pLL}+\frac{R_{d}}{L_{f}}+\frac{R_{d}}{L_{c}}) & 0\\ -w_{pLL}R_{d} & \frac{1}{C_{f}}-\frac{r_{f}R_{d}}{L_{f}} & w_{pLL}R_{d} & \frac{-1}{C_{f}}+\frac{r_{c}R_{d}}{L_{c}} & 0 & -(w_{pLL}+\frac{R_{d}}{L_{f}}+\frac{R_{d}}{L_{c}}) \end{array}\right]$$
$$B_{s}=\left[\begin{array}{cc} 1/L_{f} & 0\\ 0 & 1/L_{f}\\ 0 & 0\\ 0 & 0\\ R_{d}/L_{f} & 0\\ 0 & R_{d}/L_{f} \end{array}\right],B_{g}=\left[\begin{array}{cc} 0 & 0\\ 0 & 0\\ -1/L_{c} & 0\\ 0 & -1/L_{c}\\ R_{d}/L_{c} & 0\\ 0 & R_{d}/L_{c} \end{array}\right],C_{s}=\left[\begin{array}{cccccc} 0 & 0 & 1 & 0 & 0 & 0\\ 0 & 0 & 0 & 1 & 0 & 0\\ 0 & 0 & 0 & 0 & 1 & 0\\ 0 & 0 & 0 & 0 & 0 & 1 \end{array}\right],E_{s}=D_{s}\left[\begin{array}{cccc} 1 & 0 & 0 & 0\\ 0 & 1 & 0 & 0\\ 0 & 0 & 1 & 0\\ 0 & 0 & 0 & 1 \end{array}\right]$$
$$x_{s}={\left[I_{id},I_{iq},I_{od},I_{oq},V_{od},V_{oq}\right]}^{T},u={\left[V_{id},V_{iq}\right]}^{T},w={\left[V_{gd},V_{gq}\right]}^{T},f={\left[f_{i_{d}},f_{i_{q}},f_{v_{d}},f_{v_{q}}\right]}^{T},\xi ={\left[\xi_{i_{d}},\xi_{i_{q}},\xi_{v_{d}},\xi_{i_{d}}\right]}^{T}$$

The dq axis output voltage and current measurements are used to calculate the instantaneous active power ($P_{r}$) and reactive power (***Q***${}_{c}$) generated by the inverter.
(4)$$P_{r}=3/2\left(V_{od}I_{od}+V_{oq}I_{oq}\right)$$
(5)$$Q_{c}=3/2\left(V_{oq}I_{od}˘-V_{od}I_{oq}\right)$$
where $V_{od}$, $V_{oq}$, $I_{od}$, $I_{oq}$ are the d and q components of sensor (C.T/P.T) output voltages/currents. Instantaneous powers are then passed through low pass filters with the corner frequency $\omega_{c}$ to obtain the filtered output power.

The general state-space form of the system is:(6)$$\dot{x_{s}}=A_{s}x_{s}+B_{s}u+B_{g}w$$
(7)$$y_{s}=C_{s}x_{s}+E_{s}f+D_{s}\xi$$

The dimensions of vectors in a general form are $x_{s}\in {ℜ}^{n*1},x_{h}\in {ℜ}^{p*1},u\in {ℜ}^{m*1},$$w\in {ℜ}^{m*1},y_{s}\in {ℜ}^{p*1},f\in {ℜ}^{q*1},\xi \in {ℜ}^{q*1}$, dimensions of system matrix, actuator matrix, grid side dynamics matrix, output matrix and stable filter matrix, respectively, are $A_{s}\in {ℜ}^{n*n}$, $B_{s}\in {ℜ}^{n*m}$, $B_{g}\in {ℜ}^{n*m}$ and $C_{s}\in {ℜ}^{p*n}$ i.e., $\left[0_{p*(n-p)},I_{p*p}\right]$; whereas dimensions of fault and disturbance distribution matrices with full row and column rank are $E_{s}\in {ℜ}^{q*q}=I_{q*q}$ and $D_{s}\in {ℜ}^{q*q}=I_{q*q}$ respectively, where n≥p≥q.

Since the norm of generally any vector x is defined by
$$\left\|x\right\|≜\sqrt{x^{T}x}$$

The boundedness of fault and disturbance magnitudes is shown in terms of norm as: ‖f‖≤α and $\|\xi \|<\xi_{o}$ which along with the above-mentioned matrix dimensions are the design requirements.

### 2.3. Stable Filtering and Augmented System

Stable Filter lessens (scales down) the magnified effect of faults and disturbances on the grid system output variables. Such filters are also used to magnify the undetectably small magnitude faults. In other words, the stable filter, as used in Equation (Equation 8), is used to make the output least dependent on faults along with providing magnification of insignificantly small faults. Then, since this scaled output state variable with the information of only faulty sensors is augmented with the system states’ variable, which itself also involves the output states, but which are unscaled. In this way, the stable filtering also gives the isolation of faulty sensors, which is required for proper diagnosis process. In some cases, a stable filter can be simply a positive definite (PD) stable identity matrix with eigen-values in the left half plane, or a first order filter to suppress the high frequency noise effects in the output signal for some applications. According to the used approach, the sensor faults and disturbances appear in the output equation and the actuator faults appear in the state equation. The faulty output variables of interest are also the part of state variable, which are replicated on a stable filtered output equation but in the scaled form. So it can also be stated that the method to detect isolated faulty actuator faults is also extended for the detection of sensor faults using the SMO Theory. It also helps to replicate the techniques used earlier in [9] for actuator faults to be used for sensor faults, appearing in the output equation only.
(8)$$\dot{x_{h}}=-A_{h}x_{h}+A_{h}y_{s}$$
where $A_{h}\in {ℜ}^{p*p}$ is the stable filter matrix
(9)$$\dot{x_{h}}=-A_{h}x_{h}+A_{h}C_{s}x_{s}+A_{h}E_{s}f+A_{h}D_{s}\xi$$

For the sake of convenience and easy handling in a compact form, the system model, i.e., the states, are augmented with a stable filtered output. The augmented system is
(10)$$\left[\begin{array}{c} {\dot{x}}_{s}\\ {\dot{x}}_{h} \end{array}\right]=\left[\begin{array}{cc} A_{s} & 0\\ A_{h}C_{s} & -A_{h} \end{array}\right]\left[\begin{array}{c} x_{s}\\ x_{h} \end{array}\right]+\left[\begin{array}{cc} B_{s} & B_{g}\\ 0 & 0 \end{array}\right]\left[\begin{array}{c} u\\ w \end{array}\right]+\left[\begin{array}{c} 0\\ A_{h}E_{s} \end{array}\right]f+\left[\begin{array}{c} 0\\ A_{h}D_{s} \end{array}\right]\xi$$
(11)$${\dot{x}}_{c}=A_{c}x_{c}+B_{c}u_{c}+E_{c}f+D_{c}\xi$$
(12)$$y_{c}=C_{c}x_{c}$$
where $y_{c}\in {ℜ}^{p*1}$
(13)$$y_{c}=\left[\begin{array}{cc} 0 & I \end{array}\right]\left[\begin{array}{c} x_{s}\\ x_{h} \end{array}\right]$$
where $x_{c}=\left[\begin{array}{c} x_{s}\\ x_{h} \end{array}\right],u_{c}=\left[\begin{array}{c} u\\ w \end{array}\right]$, $A_{c}=\left[\begin{array}{cc} A_{s} & 0\\ A_{h}C_{s} & -A_{h} \end{array}\right]$, $B_{c}=\left[\begin{array}{cc} B_{s} & B_{g}\\ 0 & 0 \end{array}\right]$, $E_{c}=\left[\begin{array}{c} 0\\ A_{h}E_{s} \end{array}\right]$ and $D_{c}=\left[\begin{array}{c} 0\\ A_{h}D_{s} \end{array}\right]$.

The dimensions of the vectors are $x_{h}\in {ℜ}^{p*1}$, $x_{c}\in {ℜ}^{n_{c}*1}$, $u_{c}\in {ℜ}^{2m*1}$ whereas the dimensions of augmented system matrices are $A_{c}\in {ℜ}^{n_{c}*n_{c}}$, $B_{c}\in {ℜ}^{n_{c}*p_{c}}$, $C_{c}\in {ℜ}^{p_{c}*n_{c}}$ i.e., $\left[0_{p_{c}*\left(n_{c}-p_{c}\right)},I_{p_{c}*p_{c}}\right]$, $A_{h}E_{s}=E_{o}\in {ℜ}^{q*q}$, $A_{h}D_{s}=D_{o}\in {ℜ}^{q*q}$, where $E_{c}\in {ℜ}^{n_{c}*q_{c}}$, $D_{c}\in {ℜ}^{n_{c}*q_{c}}$ are the fault and disturbance distribution matrices, respectively, in the augmented system, such that $E_{c}=D_{c}=\left[\begin{array}{c} 0_{n*p}\\ I_{p*p} \end{array}\right]$ and for the considered augmented MG system $n_{c}=n+p=10$, $p_{c}=p=4$, $q_{c}=q=4$.

**Remark** **2.**
*As shown by [9] since $rank(C_{c}E_{c})\leq q$, the invariant zeros of $\left(A_{c},E_{c},C_{c}\right)\subseteq \lambda \left(A_{s}\right)$, so if the open loop system is stable, the system $\left(A_{c},E_{c},C_{c}\right)$ will be minimum phase. In other words, if the system has more outputs then inputs, i.e., p≥q, then it is expected that the system will not have any invariant zeros.*


## 3. Fault Diagnosis Using SMOs (Preliminaries)

The phrase ’fault diagnosis’ in literature refers to three main objectives: namely, detection, isolation, and estimation of faults in the system [27]. Since the detection of faults is not sufficient in most of the cases, instead it also requires isolation of the fault location along with an estimation of faults to manage the corrective mechanism to ensure protection of the systems. The mentioned objectives are fulfilled using methods that are categorized into four main classes: signal-based, model-based, parameter estimation-based, and observer-based approaches of fault diagnosis [28,29,30,31].

By concept and application, the observer system gets the same inputs as system i.e., (u and w) with the input distribution matrices ($B_{s},B_{g}$), system distribution applies on the estimated state and Luenberger gain ($G_{o}$) applies on stable filtered output estimation error ($e_{o}$) term along with switch term gains ($G_{m}$ and γ) also influencing the $e_{o}$ in a switch mode, and all in combination try to form the observer to replicate the system properly.

**Remark** **3**(Role of Luenberger Gain/How SMO is Different from Ordinary Luenberger Gain Observer). *This gain $G_{o}$ is the main controlling matrix parameter used to manipulate the output estimation error ($e_{o}$) to manipulate the observer system to replicate the system properly, in order to give a proper estimation of the states/outputs. The first order SMO filter is different from ordinary Luenberger observer in a way, as it involves a switch discontinuous term (ψ) that is multiplied with another gain $G_{m}$, which is also related to Luenberger gain in terms of some common matrices. The switch discontinuous term possesses another constrained internal gain (γ) and both are multiplied with stable filtered output error term ($e_{o}$). It helps the estimated states to track the actual states in a more effective way in a finite time, which is specifically termed as achieving the sliding mode. In sliding mode, a chattering path is followed (due to the discontinuous term) on the phase plane plot (i.e., the graph between $e_{o}$ and $de_{o}/dt$) to approach to zero stable filtered output estimation error. This is done by considering the SMO in terms of error system, and considering the error surface to be a sliding surface in this SMO with error surface (variable). The same SMO is used to estimate/ reconstruct the faults after having achieved the sliding surface, i.e., output estimation error to be approaching ero in the finite time, which not only ensures the reachability of error estimator SMO, but also the stability of the fault estimation process in the finite time.*

Fault Detection and State Error Estimation with SMO

SMO is used for fault diagnostics, i.e., the detection and isolation of faults by estimation of states/outputs for the considered case, i.e., MG application. The isolation property in a literal sense localizes the faulty sensors in the system; however, for the considered case, isolation and estimation are both performed by the fault estimation SMO. The standard first order SMO is given as proposed by [9,38,39], according to which the estimated states of the MG system are
(14)$$\dot{x_{o}}=A_{c}x_{o}+B_{c}u_{c}+G_{o}e_{o}+G_{m}\psi$$
where $x_{o}=\left[\begin{array}{c} x_{s}^{*}\\ x_{h}^{*} \end{array}\right]\in {ℜ}^{n_{c}*1}$ is the estimated state vector, which is an augmented vector of estimated system states $x_{s}^{*}\in {ℜ}^{n*1}$ and estimated stable filtered output $x_{h}^{*}\in {ℜ}^{p*1}$. The matrix $G_{o}\in {ℜ}^{n_{c}*p_{c}}$ is the SMO Luenberger gain of output error injection term $e_{o}\in {ℜ}^{p*1}$, which ensures that the stability of the term $\left(A_{o}=A_{c}-G_{o}C_{c}\right)$ and $G_{m}\in {ℜ}^{n_{c}*p_{c}}$ is the SMO gain of discontinuous switching term (ψ), where both $G_{m}$,$G_{m}$ need to be determined. The proposed form of (ψ) term is $\left(\psi =-\gamma \frac{P_{o}e_{o}}{\|P_{o}e_{o}\|}\right)$, where the factor (γ) is appropriately chosen as a constant gain factor depending on the application under consideration and the needs to be determined. The gain $G_{m}$ is proposed to be of the form $G_{m}=\left[\begin{array}{c} -LT^{T}\\ T^{T} \end{array}\right]$, where $T\in {ℜ}^{q*q}$ is an orthogonal matrix that can be determined by QR factorization, however, the matrices, *L* and $P_{o}$ are sub parts of a PD Lyapunov matrix P>0, which is proposed to be in the form of $P=\left[\begin{array}{cc} P_{1} & P_{1}L\\ L^{T}P_{1} & T^{T}P_{o}T+L^{T}P_{1}L \end{array}\right]>0$, where the matrices $P\in {ℜ}^{n_{c}*n_{c}},P_{1}\in {ℜ}^{n*n},P_{o}\in {ℜ}^{p*p},T\in {ℜ}^{p*p},L\in {ℜ}^{n*p}$ are to be determined [40]. The Luenberger gain $G_{o}$ is actually also determined from the Lyapunov matrix *P*, as its working will be explained in detail in the next section.

The state estimation error is determined by taking the difference of the system states as determined from the mathematical model in (11) and the estimated states based on SMO in Equation (Equation 14).
(15)$$e_{d}=e=x_{o}-x_{c}$$

The estimated output states of the system are given by
(16)$$y_{o}=C_{c}x_{o}$$
whereas the term $e_{o}$ is the output estimation error
(17)$$e_{o}=y_{o}-y_{c}$$

The output estimation error being the residual signal, which can be defined in terms of augmented error term *e*
(18)$$e_{o}=r\left(t\right)=C_{c}\left(x_{c}-x_{o}\right)=C_{c}e$$
and
$$A_{c}=\left[\begin{array}{cc} A_{11} & A_{12}\\ A_{21} & A_{22} \end{array}\right],x_{o}=\left[\begin{array}{c} {x_{s}}^{o}\\ {x_{h}}^{o} \end{array}\right]$$

Since $e_{o}$ is not the actual output error, but rather the (scaled) stable filtered output error, the empirical suggestion is to use the form of ψ in Equation (Equation 18) instead of its normalized (scaled) version for the MG application.
(19)$$\psi =-\gamma *P_{o}e_{o}$$

**Remark** **4.**
*As shown in [9], the observer, as mentioned in Equation (Equation 14), is completely insensitive to faults (f) exits if:*

*1- $Rank(C_{c}E_{c})=q$;*

*2- The invariant zeros of the system triple $\left(A_{c},E_{c},C_{c}\right)$ lie in Left Half Plane (LHP)*


Using Equation (Equation 15), the state estimation error SMO is
(20)$$\dot{e}=A_{c}e-E_{c}f-D_{c}\xi -G_{o}e_{o}+G_{m}\psi$$

Using Equation (Equation 18) its takes the form
(21)$$\dot{e}=\left(A_{c}-G_{o}C_{c}\right)e-E_{c}f-D_{c}\xi +G_{m}\psi$$
where, $A_{o}=A_{c}-G_{o}C_{c}\in {ℜ}^{n_{c}*n_{c}}$, $e\in {ℜ}^{n_{c}*1}$

The above Equations (20) or (21) are also a standard SMO ([38]), being applied on an error system for state error estimation, where the error surface is the sliding surface. The error state is further used for fault estimation. Since the error system is an augmented form of state and the stable filtered output error, i.e.,
(22)$$\left[\begin{array}{c} \dot{e_{s}}\\ \dot{e_{o}} \end{array}\right]=\left[\begin{array}{cc} A_{11} & A_{12}\\ A_{21} & A_{22} \end{array}\right]\left[\begin{array}{c} e_{s}\\ e_{o} \end{array}\right]-\left[\begin{array}{c} G_{1}\\ G_{2} \end{array}\right]e_{o}-\left[\begin{array}{c} 0\\ E_{o} \end{array}\right]f-\left[\begin{array}{c} 0\\ D_{o} \end{array}\right]\xi -\left[\begin{array}{c} LT^{T}\\ T^{T} \end{array}\right]\psi$$
where
$$e=\left[\begin{array}{c} e_{s}\\ e_{o} \end{array}\right];G_{o}=\left[\begin{array}{c} G_{1}\\ G_{2} \end{array}\right];E_{c}=\left[\begin{array}{c} 0\\ E_{o} \end{array}\right];G_{m}=\left[\begin{array}{c} LT^{T}\\ T^{T} \end{array}\right]$$

The dimensions of vectors and matrices in general form are $e_{s}\in R^{n*1}$, $e_{o}\in R^{p*1}$, $\psi \in R^{q*1}$, $f\in R^{q*1}$, $\xi \in R^{q*1}$, $A_{11}\in R^{n*n}$, $A_{12}\in R^{n*p}$, $A_{21}\in R^{p*n}$, $A_{11}\in R^{p*p}$, $D_{o}\in R^{q*q}$, $E_{o}\in R^{q*q}$, $G_{o}\in R^{n_{c}*p}$,$G_{1}\in \$R^{n*p}$, $G_{2}\in R^{p*p}$, $L\in R^{n*p}$, $T\in R^{q*q}$ whereas for the MG system considered $n=6,p=4,q=4,m=2,n_{c}=n+p=10,m_{c}=m=2,p_{c}=p=4,q_{c}=q=4$.

The next section is focused on stability analysis of the proposed observers for the MG system, as well as the following gains of fault detection and fault estimation observers, which are determined.

## 4. Stability Analysis and Determination of Robust to Disturbance/Fault Sensitive SMO Gains

**Lemma** **1.**
*For general stability analysis of SMOs, using Proposition (1) from [32], if $\left(G_{o}\right)$ is the gain of SMO for output estimation error injection term $\left(e_{o}\right)$, $G_{m}$ the SMO gain of discontinuous control term (ψ) is proposed to be of the form $G_{m}=\left[\begin{array}{c} -LT^{T}\\ T^{T} \end{array}\right]$, the constant gain (γ) of (ψ) term is constrained as ($\gamma \geq \eta_{o}-\|E_{o}\left\|\alpha_{1}\right)$ where (η>0) and P is a positive definite matrix, i.e., (P>0) of the form*

$$P=\left[\begin{array}{cc} P_{1} & P_{1}L\\ L^{T}P_{1} & T^{T}P_{o}T+L^{T}P_{1}L \end{array}\right]>0,$$

*which satisfies $\left(PA_{o}+A_{o}^{T}P<0\right)$, then the estimation error e(t) is asymptotically stable.*


**Remark** **5.**
*The Lyapunov matrix (P) is basically manipulating the energy of the error estimation system, i.e., $e^{T}e$, as P is used as the scaling matrix in the Lyapunov function. i.e., $V=e^{T}Pe$, which requires to be proved positive definite, i.e., the one with Eigen-values in the left half plane ensuring stability. Similarly, the first derivative needs to be proved to be negative or semi negative definite according to Lyapunov theory to prove its stability. The equation for dV/dt is mathematically manipulated to make the inequalities, and the vector algebraic inequality is transformed to LMIs, which are convex optimized using the MATLAB toolbox to determine the unknown design Lyapunov matrix P (with a higher degree of freedom being in the inequality form of Lyapunov equation fromwhich it is determined). It is important due to the reason that the gains of SMO ($G_{o}$ and $G_{m}$) are determined from the Lyapunov matrix (P), which ensures the proper state/output/error/fault/disturbance estimation by achieving the sliding mode required for suitable estimation process. The convex optimization tools and solvers are mentioned in Section A.4.*


Now we have to analyze the system using the criteria of robustness to disturbance H∞, the criteria of sensitivity to faults H−, and the compromised criteria H−/H∞ for fault diagnosis and tolerance analysis. The theory depends more on Game Theoretic estimation being utilized in Hamilton–Jacobi–Isaacs-Equation (HJIE) and to convert the equalities to inequalities for better handling and more design freedom.

### 4.1. H∞ Robustness Analysis

**Lemma** **2.**
*According to Problem 3.1 in [27], for the system defined in Equation (Equation 11), the maximum robustness attenuation problem*

(23)
$$H\infty =sup_{\left[f=0,\xi \neq 0\right]}\frac{\|r_{K}{\|}_{2,[0,t]}}{{\left\|\xi \right\|}_{2,[0,t]}}\leq \alpha$$

*is satisfied, if the cost functional*

(24)
$$H_{o}\left(G_{o},\xi \right)=\int \left({r_{K}}^{T}r_{K}-\alpha^{2}\xi^{T}\xi \right)dt;\left(for\right)f=0$$

*is smaller than or equal to zero for any possible disturbance, where (α) is (H∞) parameter and residual signal $\left(r_{K}=Kr\left(t\right)=KC_{c}e\right)$ in this case is state estimation error. The problem is now viewed/explained as two player zero sum differential game with the above defined cost functional in Equation (Equation 24), where the maximizing player tries to maximize the functional through ξ and the minimizing player minimizes the functional through ($G_{o},G_{m}$). Then, using the concepts of dynamic game theory, the cost function in Equation (Equation 24) gives the pair of strategies $\left(G_{o}^{*},\xi^{*}\right)$, providing a saddle-point solution*

(25)
$$H_{o}\left(G_{o},\xi^{*}\right)\leq H_{o}\left(G_{o}^{*},\xi^{*}\right)\leq H_{o}\left(G_{o}^{*},\xi \right)$$



**Definition** **2**(Saddle Point). *In mathematics, a minimax point or saddle point is a point on the surface of the graph of a function where the slopes (derivatives) in the orthogonal directions are all zero (a critical point), but which is not a local extremum of the function. The saddle point of the problem is taken in terms of cost functionals.*

Its mathematical details are shifted in the Appendix A.

**Lemma** **3.**
*Considering the cost functional of disturbance attenuation problem from Equation (Equation 24), constraining $\left(H_{o}\left(G_{o},\xi \right)<0\right)$ and using the definition, for any general state $x^{*}$*

$$\frac{dV_{1}\left(x^{*},t\right)}{dt}=\frac{\partial V_{1}\left(x^{*},t\right)}{\partial t}+\frac{\partial V_{1}\left(x^{*},t\right)}{\partial x^{*}}\frac{\partial x^{*}}{\partial t}$$

*The inequality version of HJI equation , as mentioned in Equation (A4) in Lemma A1 of Section A.5, is*

(26)
$$\frac{-\partial V_{1}\left(x^{*}\left(t\right),t\right)}{\partial t}\leq \frac{\partial V_{1}\left(x^{*},t\right)}{\partial x^{*}}\dot{x^{*}}+{r_{K}}^{T}r_{K}-\alpha^{2}\xi^{T}\xi$$



**Remark** **6.**
*The derivation for H∞ constrained inequality version of HJIE is given in Lemma A3 in Section A.5. The problem can be studied in detail in [27,28,29].*


**Remark** **7.**
*Considering Section A.5, using the HJIE in Equations (A4) and (A7), the Hamiltonians in Equations (A5) and (A7) and the disturbance attenuation/fault sensitivity constraints in Equations (24) and (35), the analytical solution for gain $G_{o}$ will be dependent on states that are undesired (theoretically by Luenberger linear observer theory). However, the inequality version of HJIE will give more freedom in choosing the Lyapunov function $V_{1}\left(e,t\right)$ and hence more freedom in the design of the sliding mode observer gain $G_{o}$ being state independent.*


**Remark** **8.**
*Approach of Using HJIE, H∞, H− and Game Theory in This Study:*

*The approach used here is to design the observers for linear/non-linear systems based on game theoretic saddle point estimation. The H∞ and H− parameters deal with the extreme cases in a way that they provide robustness to worst case disturbances and sensitivity to minimum faults. These parameters in inequality form are similar in nature to the saddle point of the game theory, as described by Equations (25) and (36). The H∞ and H− constraints are also part of the Hamiltonians (in Equations (A5) and (A7) in Section A.5) along with the Lyapunov (energy) function. According to the approach used by [27], the Hamiltonian and its derivative w.r.t faults (f)/disturbances (ξ)/SMO gain ($G_{o}$) following the H∞ and H− constraints can let us determine the optimal values of faults/disturbances, i.e., ($f^{*}$, $\xi^{*}$, $G_{o}$), by its minimization. However, this work instead uses the approach that the H∞ constraint in inequality form, inspired by the saddle point, is manipulated to form the inequality version of HJIE, which has the Hamiltonian incorporated in it as well.*


The resulting HJIE consists of the faults/disturbances, Lyapunov function, and output estimation error ($e_{o}$) as variables. The inequality version of this HJIE not only ensures Lyapunov stability but also gives the LMIs, which are convex and optimized using the MATLAB toolbox to find the optimized SMO gains, which are used to estimate the states/outputs/state errors.

The game theoretic saddle point’s approach for incorporating the H∞ and H− norms also has the application for the considered faulty and perturbed systems due to the reason that faults and disturbances are unknown. In this case the worst case fault and disturbance values are used for observer/filter designs for robust residual generation and fault/disturbance estimations. The SMO in reduced order is used to reconstruct faults/disturbances, while following the constraints of fault-detection-sensitivity and rejecting the effects of disturbances through the observer gains. The proof for HJIE and some more details are given in the appendix in Section A.3.

**Theorem** **1.**
*If $V_{1}$ defines the positive definite Lyapunov function which satisfies the HJIE in Equation (Equation 26) constrained with disturbance attenuation problem defined in Equation (Equation 24), then the Lyapunov function in vector form gives*

*LMI $L_{RD}=\left[\begin{array}{ccc} \left(A_{c}-G_{o}C_{c}\right)Q+Q\left(A_{c}-G_{o}C_{c}\right)+C_{a}^{T}FC_{a} & -3QE_{c} & -QD_{c}\\ -E_{c}^{T}Q & 0 & 0\\ -D_{c}^{T}Q & 0 & -\alpha^{2}I \end{array}\right]\leq 0$*
*which is a convex optimized iteratively to give a robust set value of the worst case disturbance SMO gain defined by $G_{o}=Q^{-1}C_{c}^{T}{F^{\prime }}^{-1}$.*


**Proof.** Considering the general HJIE in Equation (Equation 26) with the cost functional in Equation (Equation 24) to ensure maximum robustness to a worst case disturbance
$$\frac{\partial V_{1}\left(e\left(t\right),t\right)}{\partial t}+\frac{\partial V_{1}\left(e\left(t\right),t\right)}{\partial e}\dot{e}+{r_{K}}^{T}r_{K}-\alpha^{2}\xi^{T}\xi \leq 0$$Since $\left(V_{1}=e^{T}Qe\right)$ and $\left(\frac{\partial V_{1}\left(e,t\right)}{\partial t}={\dot{V}}_{1}\left(e,t\right)=\dot{e^{T}}Qe+e^{T}Q\dot{e}\right)$Using the above given H∞ constrained HJIE equation
(27)$$\dot{e^{T}}Qe+e^{T}Q\dot{e}+\left(2\dot{e^{T}}\right)Qe-\alpha^{2}\xi^{T}\xi +e^{T}C_{c}^{T}K^{T}KC_{c}e\leq 0$$
where $K^{T}K=F^{\prime }=\left[\begin{array}{c} 0\\ F \end{array}\right]$ and the matrix $F\in R^{p*p}$ as the sub part of matrix *K*
(28)$$\dot{e^{T}}Qe+3e^{T}Q\dot{e}-\alpha^{2}\xi^{T}\xi +e^{T}C_{c}^{T}F^{\prime }C_{c}e\leq 0$$Substituting equations for $\dot{e}$
$$e^{T}{\left(A_{c}-G_{o}C_{c}\right)}^{T}Qe+3e^{T}Q\left(A_{c}-G_{o}C_{c}\right)e-f^{T}E_{c}^{T}Pe-\xi^{T}D_{c}^{T}Qe+\psi G_{n}^{T}Qe$$
(29)$$-3e^{T}QE_{c}f-3e^{T}QD_{c}\xi +3e^{T}QG_{n}\psi +e^{T}C_{c}^{T}F^{\prime }C_{c}e-\alpha^{2}\xi^{T}\xi \leq 0$$The inequality in vector form gives:
(30)$$\left[\begin{array}{ccc} e^{T} & f^{T} & \xi^{T} \end{array}\right]\left[\begin{array}{ccc} \left(A_{c}-G_{o}C_{c}\right)Q+Q\left(A_{c}-G_{o}C_{c}\right)+C_{c}^{T}F^{\prime }C_{c} & 3QE_{c} & -QD_{c}\\ -E_{c}^{T}Q & 0 & 0\\ -D_{c}^{T}Q & 0 & -\alpha^{2}I \end{array}\right]\left[\begin{array}{c} e\\ f\\ \xi \end{array}\right]\leq 0$$The LMI obtained from the vector Lyapunov equation is
(31)$$L_{RD}=\left[\begin{array}{ccc} \left(A_{c}-G_{o}C_{c}\right)Q+P\left(A_{c}-G_{o}C_{c}\right)+C_{c}^{T}F^{\prime }C_{c} & 3QE_{c} & -QD_{c}\\ -E_{c}^{T}Q & 0 & 0\\ -D_{c}^{T}Q & 0 & -\alpha^{2}I \end{array}\right]\leq 0$$As according to the cost functional in Equation (Equation 24) f=0, dropping the disturbance terms gives the LMI
(32)$$\left[\begin{array}{cc} \left(A_{c}-G_{o}C_{c}\right)Q+Q\left(A_{c}-G_{o}C_{c}\right)+C_{c}^{T}F^{\prime }C_{c} & -QD_{c}\\ -D_{c}^{T}Q & -\alpha^{2}I \end{array}\right]\leq 0$$□

**Remark** **9.**
*The LMIs obtained using the above process are processed further with an algebraic Riccati equation according to Theorem (2) in [32] to avoid optimization infeasibility issues with (H∞) criterion LMIs, specific to the system. The definition and a brief mathematical explanation of the Riccati equation is given in Definition A1 in Section A.6*

*The modified LMI is thus given by*

(33)
$$L_{RDO}=\left[\begin{array}{cc} A_{c}Q+QA_{c}-3YC_{c} & -QD_{c}\\ -D_{c}^{T}Q & -3\alpha^{2}I \end{array}\right]\leq 0$$


*where $Y=QG_{o}$*
*The LMI is solved by iterative convex optimization to determine the robust to disturbance sliding mode observer gains. The tools and solvers used are mentioned in Section A.4*


### 4.2. H−Minimum Fault Sensitivity Analysis

**Lemma** **4.**
*According to Problem 3.2 in [27], for the system defined in Equation (Equation 11), the maximum sensitivity to minimum fault problem*

(34)
$$inf_{\left[\xi =0,f\neq 0\right]}\frac{\|r_{K}{\|}_{2,[0,t]}}{{\left\|f\right\|}_{2,[0,t]}}\geq \beta^{2}$$

*is satisfied, if the cost functional*

(35)
$$H_{o}\left(G_{o},f\right)=\int \left({r_{K}}^{T}r_{K}-\beta f^{T}f\right)dt;\left(\xi =0\right)$$

*is greater than or equal to zero for each possible fault. This can be viewed as a two player zero-sum differential game with the cost functional. The minimizing player tries to minimize the functional through f and the maximizing player maximizes the functional through $G_{o}$. Then, using the concepts of dynamic game theory, the cost functional in Equation (Equation 35) gives the pair of strategies $\left(H^{*},f^{*}\right)$, providing a saddle-point solution, i.e.,*

(36)
$$H_{o}\left(G_{o},f^{*}\right)\leq H_{o}\left(G_{o}^{*},f^{*}\right)\leq H_{o}\left(G_{o}^{*},f\right)$$


*Its mathematical detail is also shifted in the appendix .*


**Lemma** **5.**
*Considering the cost functional of fault sensitivity problem from Equation (Equation 35), constraining $\left(H_{o}\left(G_{o},f\right)<0\right)$ and using the definition, for any general state $x^{*}$*

$$\frac{dV_{2}\left(x^{*},t\right)}{dt}=\frac{\partial V_{2}\left(x^{*},t\right)}{\partial t}+\frac{\partial V_{2}\left(x^{*},t\right)}{\partial x^{*}}\frac{\partial x^{*}}{\partial t}$$

*The inequality version of the HJIE Equation in (37) is*

(37)
$$\frac{-\partial V_{2}\left(e\left(t\right),t\right)}{\partial t}\leq \frac{\partial V_{2}\left(e\left(t\right),t\right)}{\partial e}\dot{e}+{r_{K}}^{T}r_{K}-\beta^{2}f^{T}f$$



**Remark** **10.**
*The mathematical proof for the H− constrained inequality version of HJIE is given in Lemma A4 in Section A.5. The problem can be studied in detail in [27,28,29].*


**Theorem** **2.**
*If $V_{2}$ defines the positive definite Lyapunov function that satisfies the HJIE in Equation (Equation 37) constrained with the maximum sensitivity of the minimum set fault problem defined in Equation (Equation 35), then the Lyapunov function in vector form gives LMI*
*$L_{SF}=\left[\begin{array}{ccc} -\left(A_{c}-G_{o}C_{c}\right)P-P\left(A_{c}-G_{o}C_{c}\right)-C_{c}^{T}F^{\prime }C_{c} & 3PE_{c} & PD_{c}\\ E_{c}^{T}P & \beta^{2}I & 0\\ D_{c}^{T}P & 0 & 0 \end{array}\right]\leq 0$**which in a reduced form is a convex optimized iteratively to give a sensitive set value of minimum case fault SMO gain defined by $G_{o}=P^{-1}C_{c}^{T}{F^{\prime }}^{-1}$*.

**Proof.** Considering the general HJIE in Equation (Equation 37) with the cost functional in Equation (Equation 35) to ensure maximum sensitivity to minimum case fault
$$\frac{-\partial V_{2}\left(e\left(t\right),t\right)}{\partial t}\leq \frac{\partial V_{2}\left(e\left(t\right),t\right)}{\partial e}\dot{e}+{r_{K}}^{T}r_{K}-\beta^{2}f^{T}f$$
Since $\left(V_{2}=e^{T}Pe\right)$ and $\left(\frac{\partial V_{2}\left(e,t\right)}{\partial t}={\dot{V}}_{2}\left(e,t\right)=\dot{e^{T}}Pe+e^{T}P\dot{e}\right)$Using the above given H− constrained HJIE equation
(38)$$\dot{e^{T}}Pe+e^{T}P\dot{e}\leq \left(2\dot{e^{T}}\right)Pe-\beta^{2}f^{T}f+e^{T}C_{c}^{T}K^{T}KC_{c}e$$
(39)$$\dot{e^{T}}Pe+3e^{T}P\dot{e}+e^{T}C_{c}^{T}F^{\prime }C_{c}e-\beta^{2}f^{T}f\geq 0$$
(40)$$-\dot{e^{T}}Pe-3e^{T}P\dot{e}-e^{T}C_{c}^{T}F^{\prime }C_{c}e+\beta^{2}f^{T}f\leq 0$$Substituting equations for $\dot{e}$
$$3e^{T}\left(A_{c}-G_{o}C_{c}\right)Pe+3e^{T}P\left(A_{c}-G_{o}C_{c}\right)e-f^{T}E_{c}^{T}Pe-\xi^{T}D_{c}^{T}Pe+\psi G_{n}^{T}Pe-$$
(41)$$3e^{T}PE_{c}f-3e^{T}PD_{c}\xi +3e^{T}PG_{n}\psi +e^{T}C_{c}^{T}F^{\prime }C_{c}e-\beta^{2}f^{T}f\leq 0$$The inequality in vector form gives
(42)$$\left[\begin{array}{ccc} e^{T} & f^{T} & \xi^{T} \end{array}\right]\left[\begin{array}{ccc} -\left(A_{c}-G_{o}C_{c}\right)P-P\left(A_{c}-G_{o}C_{c}\right)-C_{c}^{T}F^{\prime }C_{c} & 3PE_{c} & PD_{c}\\ E_{c}^{T}P & \beta^{2}I & 0\\ D_{c}^{T}P & 0 & 0 \end{array}\right]\left[\begin{array}{c} e\\ f\\ \xi \end{array}\right]\leq 0$$The LMI obtained from the vector Lyapunov equation is
(43)$$L_{SF}=\left[\begin{array}{ccc} -\left(A_{c}-G_{o}C_{c}\right)P-P\left(A_{c}-G_{o}C_{c}\right)-C_{c}^{T}F^{\prime }C_{c} & 3PE_{c} & PD_{c}\\ E_{c}^{T}P & \beta^{2}I & 0\\ D_{c}^{T}P & 0 & 0 \end{array}\right]\leq 0$$According to the cost functional in Equation (Equation 35) ξ=0, then dropping the disturbance terms makes the LMI optimized
(44)$$L_{SFO}=\left[\begin{array}{cc} -\left(A_{c}-G_{o}C_{c}\right)P-P\left(A_{c}-G_{o}C_{c}\right)-C_{c}^{T}F^{\prime }C_{c} & PE_{c}\\ E_{c}^{T}P & \beta^{2}I \end{array}\right]\leq 0$$The LMI is solved by iterative convex optimization to determine the fault sensitive sliding mode observer gains. □

### 4.3. Theorem 3: H−/H∞ Criteria Based on HJIE for Observer Design

If $V_{1}=e^{T}Pe$ and $V_{2}=e^{T}Qe$ define the positive definite Lyapunov functions that satisfy the HJIEs in Equations (26) and (37) constrained with disturbance attenuation and minimum sensitivity problems defined in Equations (24) and (35), respectively, then the Lyapunov functions in vector form gives LMIs

$L_{SF}=\left[\begin{array}{ccc} -\left(A_{c}-G_{o}C_{c}\right)P-P\left(A_{c}-G_{o}C_{c}\right)-C_{c}^{T}F^{\prime }C_{c} & 3PE_{c} & PD_{c}\\ E_{c}^{T}P & \beta^{2}I & 0\\ D_{c}^{T}P & 0 & 0 \end{array}\right]\leq 0$
and $L_{RD}=\left[\begin{array}{ccc} \left(A_{c}-G_{o}C_{c}\right)Q+Q\left(A_{c}-G_{o}C_{c}\right)+C_{c}^{T}F^{\prime }C_{c} & 3QE_{c} & -QD_{c}\\ -E_{c}^{T}Q & 0 & 0\\ -D_{c}^{T}Q & 0 & \alpha^{2}I \end{array}\right]\leq 0$ which in a modified and reduced form are a convex optimized iteratively as a mixed problem to give a compromised form of SMO gains $G_{o}$ and $G_{m}$, which possesses robustness to worst case disturbance along with sensitivity to minimum fault at the same time. The Luenberger gain $G_{o}$ is defined by $G_{o}=P^{-1}C_{c}^{T}{F^{\prime }}^{-1}$ in Theorem 1.

**Proof.** Considering the general HJIEs in Equations (26) and (37) with cost functions in Equations (24) and (35) to ensure maximum robustness to worst case disturbance and sensitivity to minimum fault at the same timeLet, if the Lyapunov functions in terms of positive definite matrices, (P>0) and (Q>0) be $V_{1}=e^{T}Pe$ and $V_{2}=e^{T}Qe.$Using the H− constrained HJIE inequality version (37) and vector form from Equation (Equation 43)

$-3e^{T}\left(A_{c}-G_{o}C_{c}\right)Pe-3e^{T}P\left(A_{c}-G_{o}C_{c}\right)e+f^{T}E_{c}^{T}Pe+\xi^{T}D_{c}^{T}Pe-\psi G_{n}^{T}Pe-3e^{T}PE_{c}f-3e^{T}PD_{c}\xi -3e^{T}PG_{n}\psi -e^{T}C_{c}^{T}F^{\prime }C_{c}e+\beta^{2}f^{T}f\leq 0$


$$\left[\begin{array}{ccc} e^{T} & f^{T} & \xi^{T} \end{array}\right]\left[\begin{array}{ccc} -\left(A_{c}-G_{o}C_{c}\right)P-P\left(A_{c}-G_{o}C_{c}\right)-C_{c}^{T}F^{\prime }C_{c} & 3PE_{c} & PD_{c}\\ E_{c}^{T}P & \beta^{2}I & 0\\ D_{c}^{T}P & 0 & 0 \end{array}\right]\left[\begin{array}{c} e\\ f\\ \xi \end{array}\right]\leq 0$$

Using the H∞ constrained HJIE inequality version (26) and vector form from Equation (Equation 31)$e^{T}{\left(A_{c}-G_{o}C_{c}\right)}^{T}Qe+3e^{T}Q\left(A_{c}-G_{o}C_{c}\right)e-f^{T}E_{c}^{T}Pe-\xi^{T}D_{c}^{T}Qe+\psi G_{n}^{T}Qe+3e^{T}PE_{c}f-3e^{T}PD_{c}\xi +3e^{T}PG_{n}\psi +e^{T}C_{c}^{T}F^{\prime }C_{c}e-\alpha^{2}\xi^{T}\xi \leq 0$$$\left[\begin{array}{ccc} e^{T} & f^{T} & \xi^{T} \end{array}\right]\left[\begin{array}{ccc} \left(A_{c}-G_{o}C_{c}\right)Q+Q\left(A_{c}-G_{o}C_{c}\right)+C_{c}^{T}F^{\prime }C_{c} & -3QE_{c} & -QD_{c}\\ -E_{c}^{T}Q & 0 & 0\\ -D_{c}^{T}Q & 0 & -\alpha^{2}I \end{array}\right]\left[\begin{array}{c} e\\ f\\ \xi \end{array}\right]\leq 0$$
the LMIs obtained for the mixed H−/H∞ problem are the same as given by Equations (44) and (33)
$$L_{SF}=\left[\begin{array}{ccc} -\left(A_{c}-G_{o}C_{c}\right)P-P\left(A_{c}-G_{o}C_{c}\right)-C_{c}^{T}F^{\prime }C_{c} & 3PE_{c} & PD_{c}\\ E_{c}^{T}P & \beta^{2}I & 0\\ D_{c}^{T}P & 0 & 0 \end{array}\right]\leq 0$$
$$L_{RD}=\left[\begin{array}{ccc} \left(A_{c}-G_{o}C_{c}\right)Q+Q\left(A_{c}-G_{o}C_{c}\right)+C_{c}^{T}F^{\prime }C_{c} & -3QE_{c} & -QD_{c}\\ -E_{c}^{T}Q & 0 & 0\\ -D_{c}^{T}Q & 0 & -\alpha^{2}I \end{array}\right]\leq 0$$The LMIs to be optimized in reduced and modified forms are given in Equations (33) and (44) to give gains of SMOs following the mixed constraint, i.e., are sensitive to faults and robust to disturbance at the same time. □

**Remark** **11.**
*According to Algorithm 5.1.1 in [29] for the mixed problem with H−/H∞ constraint, both inequalities are taken either greater than zero or less than zero with signs of terms being reversed for one of the constraints according to the HJIE equation, so that LMI optimization stays possible along with a condition on Lyapunov matrices as P=Q.*


**Remark** **12.**
*The algorithm used for optimization is in [29], which mainly imposes the condition on Lyapunov matrices P=Q.*


**Lemma** **6.**
*Using Theorem 3 in [32] if the state estimation error system (22) is transformed by transformation matrix*

(45)
$$T_{L}=\left[\begin{array}{cc} I_{n-p} & L\\ 0 & T \end{array}\right]$$

*to induce reduced order sliding motion on estimation SMO, the design Lyapunov matrix is constrained as $\left(P_{o}T\left(A_{22}-G_{2}\right)+\left(A_{22}^{T}-G_{2}^{T}\right)T^{T}P_{o}<0\right)$ and if reduced order sliding motion is governed by $\sigma \left(e_{o}\right)=\left\{e_{o}:Ce_{o}=0\right\}$, such that the fault and disturbance magnitude is bounded, i.e., $\left(\|f\|<f_{o}\right)$ and $\left(\|\xi \|\leq \xi_{o}\right)$ and the gain factor (γ) for output error injection term is bounded by $\left(\gamma \geq \|T{\bar{A}}_{21}\|\|e_{s}\|-\|TE_{o}\|\alpha_{o}-\|TD_{o}\|\xi_{o}+\eta \right)$, then it will be ensured that the fault detection and estimation SMOs utilized for MG system are completely stable in terms of Lyapunov criteria and ensure finite time reachability of sliding motion. The finite time of reachability to sliding surface is given by*

(46)
$$T_{R}\leq \sqrt{\frac{e_{o}^{T}P_{o}e_{o}}{\lambda_{min}\left(P_{o}^{-1}\right)}}$$



**Lemma** **7.**
*Using Corollary 1 in [32], considering the MG system in Equation (Equation 11) and the observer system in Equation (Equation 14), if the LMI optimization matrix described in Theorems 1 or 2 is solved to determine SMO gains, the error system in Equation (Equation 22) after being transformed by matrix in Equation (Equation 45) gives reduced order error system,, and if the stability of observers is ensured by constant gain of output injection term γ, which is constrained to satisfy $\left(\gamma \geq \|T{\bar{A}}_{21}\|\|e_{s}\|-\|TE_{o}\|\alpha -\|TD_{o}\|\xi_{o}+\eta \right)$, where β is a scaling constant, then the sensor fault $\left(f^{*}\right)$ and disturbance $\left(\xi^{*}\right)$ can be estimated as*

(47)
$$f^{*}=S_{f}E_{o}^{-1}T^{-1}\psi_{eq}$$


(48)
$$\xi^{*}=f-f^{*}-E_{o}^{-1}T^{-1}A_{21}e_{s}$$

*where $S_{f}$ is a fault scaling constant.*


**Remark** **13.**
*Lemmas 6 and 7 are based on the base paper by the same authors [32].*


**Comment 1:** The FTC approach is a requirement of simulation, and hence it is part of the paper along with diagnosis analysis, but its not the part of this investigation. Therefore, it is also used in a similar way as in [32]. The simulation result is included in the results.

## 5. FTC Approach/Working

The estimated faults by using SMOs are added/subtracted from the faulty sensor readings and the corrected values are fed to the PI based control block, which is also working for the common microgrid problems, i.e., balancing the voltage and frequency sags. However, this work does not contribute to the control block, except to provide corrections in faulty sensor readings by using robust and fault sensitive faults/ disturbances estimators. The accurate fault estimation/reconstruction will provide the correction closest to the actual one, which is added/subtracted to the faulty sensor readings to get the actual one. So, the fault tolerance surely depends upon accurate fault estimation and least fault estimation error in the results and discussion section. The estimated fault is subtracted from the faulty sensors’ output vector.
(49)$$Y_{c}=Y_{s}-E*f^{*}$$
where $f^{*}$ is the estimated fault and $Y_{u}$ is the vector of faulty sensor outputs, and $Y_{c}$ is the corrected sensor outputs vector. The complete working of the system and proposed solution is shown in the Simulink-based detailed block diagram in Figure 3.

**Remark** **14.**
*The above equation shows that the working and effectiveness of fault tolerance depends on accurate fault estimation. The accurateness of fault estimation can be observed in the fault estimation errors with fault estimation performed with various constrained SMOs. The fault estimation error graphs for voltage/currents are given in figures in the results and discussions section.*


**Remark** **15.**
*The contribution of this work is not to consider the voltage and frequency sags/droops occurring in the islanded/grid-connected microgrid, but instead we have considered the faults of C.T/P.T, which are rectified using the SMOs for fault detection and estimation. The estimated faults are given as corrections of the faulty sensor readings, and in this way the software based sensor serves as a replacement sensor and the corrected readings are used by the control block to serve as a controller for voltage-frequency sags as well.*


The procedure for complete observer based FTC approach as given by Figure 3 is explained in Algorithm 1.
**Algorithm 1.** Algorithm of the Procedure.Inputs: u, w, f, ξOutputs: Matrix P, Gains ($G_{o},G_{m}$), $x_{s}$, $y_{s}$, $x_{o}$, $y_{o}$, e, $e_{o}$, γ, $\Psi_{\mathrm{eq}}$, $e_{s}$, $f^{*}$, $\xi^{*}$while (For the given Time of simulation/process)START:         **1:** d-q-0 transform the linearized system model Equation (2) by Equation (3);         **2:** Take converter/grid voltages (u, w) from Simulink hard wired microgrid model in Figure 2;         **3:** Generate the faults and disturbances to be added (representing) sensor/C.T/P.T faults;         **4:** Give voltages (u, w) as inputs to the Simulink-based mathematical model of a microgrid, in Equation (6);         **5:** Add faults/disturbances (f, ξ) to the output equation of the system represented by Equation (7);         **6:** Pass the faulty system output in Equation (6) from a stable filter defined in Equations (8) and (9) to reduce the magnified effect of faults and disturbances;         **7:** Augment the system states with stable filtered output, as done in Equations (10) and (11);         **8:** Pass the augmented system state from state/output estimator SMO in Equation (14);         **9:** Determine the augmented (state/stable filtered output) estimation error by taking the difference between system and observer states/outputs as defined in Equation (15);         **10:** Use H− constraint mentioned in Equation (35) (β parameter is worst case/ minimal fault in Equation (34)) in HJIE in Equation (37) (to fulfill the saddle point requirement);         <H∞ constraint in parallel is given by Equation (23), the worst case disturbance upper bound in Equation (24), and the respective HJIE in Equation (26)>;         **11:** Determine the constrained LMIs from the above equation in vector form as defined in (44) for H− and Equation (32) for H∞;         **12:** Convex feasibility and optimize the constrained LMI given in Equation (44) (for H− constraint) and Equation (32) (for H∞ constraint) using the function ‘feasp’ in the Section A.4 to find the optimal gain $G_{o}^{*}$ as defined in the Theorem 1 statement, and switch term gain $G_{m}^{*}$, as defined in Lemma 1; <Gains are determined from Lyapunov P or Q matrices , using Lemma (1) and given in detail in [32] >;         **13:** Use the same SMO gains in state/output estimator SMO in step 8;         **14:** Determine the state estimation error as determined in Equation (15);         **15:** Give stable filtered output estimation error part of the total error vector (e), i.e., ($e_{o}$) in Equations (17) and (18) to state/output estimator SMO in step 8 and the state error estimator SMO in Equations (20) and (21) to attain the sliding mode;         **16:** If (the sliding mode is attained in Equations (20) or (21)),Feed the state estimation error to the reduced order state estimation error, explained in Lemma 7 (details in [32]);
         **17:** Determine the gains (γ (defined in Lemma 7), $\Psi_{\mathrm{eq}}$, reduced order state error $e_{s}$)) (details in [32]);         **18:** Compute the estimated fault as done in Equation (47) and the disturbance as in Equation (48);         **19:** Use the estimated faults and correct the faulty sensor readings by adding/ subtracting from it, as done in Equation (49).         **20:** Feed the corrected sensor output values to PI and Droop based current/voltage/real power/complex power control as shown in the detailed Simulink based block diagram with details in the FTC section of [32];         **21:** Repeat Step 10 onward for the H∞ constraint in (24) and the compromised H−/H∞ constraints.END (of while loop)

## 6. Results and Discussions

The simulations are performed for state/output estimation error, fault estimation/ reconstruction, fault estimation error, and fault tolerant control (FTC) performance. The behavior of the system and results for all simulations are consistent and technically no statistical analysis is required for a deterministic system and simulation platform.

The Simulink-based detailed three phased inverter model is considered in the simulation. The DC voltage and grid voltage are both working at 600 V. PLL block and all abc-dq transformations are referencing the phase of grid voltage phase; SVPWM uses a frequency of 10,000 Hz and a sampling time of 0.0001; however, the default internal settings takes the samples at 0.0002 s. The greater sampling times cause discontinuities, which can be reduced to improve the accuracy at the cost of increased response time and lesser ability of online working. Since the continuous time simulations halt or move at very low speeds, which are not viable for real time online performance, Fixed Step solvers are therefore used for simulation in Simulink (Matlab) with single task handling to avoid complexities with very minor compromise on accuracy.

Regarding some other simulation constants, for the considered time of simulation, as an estimate using the minimum/maximum values of faults, disturbances and stable filtered output error, the H∞ constant $\alpha =3.76\times 10^{-9}$ and H− constant $\beta =4.44\times 10^{-19}$.

The value of η is a small positive constant considered η=10 to ensure the constraint in inequality used in Lemma A1, whereas the upper bounds for current and voltage is considered up to $f_{o}$ = 10*A*/100*V* for I/V, respectively; whereas the upper bounds of $\xi_{o}$ are 3*A*/10*V* for I/V.

The grid parameters are given in Table 1.

A detailed Simulink-based simulation diagram is given in Figure 3. From the top right, the block named ‘Microgrid System’ is the wired microgrid system, as shown in Figure 2 in Section 2. The inverter and grid input voltages are given to the system by this block. The second block, named the ’Voltage Frequency Control’ block, manages the voltage/frequency sags of corrected voltage/current signals. The third block, named ’Fault/Disturbance Injection in sensor Outputs’, injects the faults and disturbances according to the fault model in Equation (Equation 1) in a block named ’Simulink Based System Model’ (which is mentioned in Equations (6) and (7)) as well as in a block named ’Augmented System Model and Stable Filtering’ (as done in Equations (10) or (11)). The faulty system outputs are given to the blocks named ’FD Block/State and Output Estimation SMO’ (defined by Equation (Equation 14)) and ’Error (State/Output) Estimation SMO’ (as defined in (18)). These outputs are given to the block named ‘Reduced Order state Error’, which computes value of γ gain, values of switched signal in normal and sliding mode (i.e., ψ, $\psi_{\mathrm{eq}}$), and reduced order state estimation error ($e_{s}$) (which are the parameters required by the equations mentioned in Lemma 7), while these computed values are also given back to ’FD Block’ and ’Error Estimation SMO’ particularly required by switch gain term of SMOs. these values are then provided to block named ’Estimated Faults/Disturbances Computation’ to compute the faults/disturbances (unknown inputs). The last block, named ‘Data Acquisition’, performs data acquisition of all required data to be logged or provided to m-files required for plotting the required results. The gains are not updated on the run time since that requires GPU based systems. However, the system can easily incorporate the run time computation of gains. The SMOs and other blocks are using the optimized constant gains computed by matlab m-files using LMIs’ convex optimization tool box commands.

Figure 4 shows reconstruction of voltage fault compared using SMO with gains optimized with fault sensitivity parameter (H−), robustness to disturbance parameter (H∞), and a mixed (H−/H∞). The voltage fault reconstruction using (H−) and (H−/H∞ are similar and nearly accurate, whereas that with (H∞) is though to be accurate but lagged with a phase of π radian. The constant multiple $S_{f}$ used in Equation (Equation 47) for voltage fault reconstruction for (H∞), (H−) and (H−/H∞) are $4\times 10^{3}$, $0.5\times 10^{14}$ and $7\times 10^{11}$, respectively. The multiples are required to compensate for the stable filtering scaling effect in output error term $e_{o}$ and are in direct relation with multiple of Luenberger gain $G_{o}$ used in SMO.

Figure 5 shows reconstruction of the current fault compared using SMO with gains optimized with fault sensitivity parameter (H−), robustness to disturbance parameter (H∞), and a mixed (H−/H∞). The current fault reconstruction using (H−) is nearly accurate, while using (H−/H∞ is relatively accurate in one half cycle and inaccurate in another half cycle, whereas the result with (H−∞) is though to be accurate but lagged with a phase of π radian, which can be corrected using the phase compensation of π radian. The constant multiple $S_{f}$ used in Equation (Equation 51) for current fault reconstruction for (H∞), (H−) and (H−/H∞) are $2.5\times 10^{3}$, $1+0.43\times 10^{14}$ and $1+3\times 10^{11}$, respectively.

Figure 6 shows voltage fault estimation errors compared using SMO with gains optimized with fault sensitivity parameter (H−), robustness to disturbance parameter (H∞), and a mixed (H−/H∞). The error is harmonic sinusoidal variation, which approaches the peak value of nearly 20 V for a very small instance of time using (H−) and (H−/H∞, being quite similar, whereas that with (H∞) is nearly a pure sinusoidal variation of 60 Hz that approaches the peak value of 175 V due to phase lag of the π radian.

Figure 7 shows current fault estimation errors compared using SMO with gains optimized with fault sensitivity parameter (H−) and robustness to disturbance parameter (H∞). However, for /H∞ and mixed (H−/H∞) parameters, the peaks approach 20 A and 11 A, respectively, which is relatively higher and impractical. The error for could be reduced to minimal if phase compensation of π radian is used in the estimated faults.

Figure 8 shows state estimation errors compared using SMO with gains optimized with robustness to disturbance parameter (H∞). The state estimation error is a sinusoidal variation with line frequency with the peak magnitudes of $V_{o_{d}q}$ and $I_{o_{d}q}$ being 0.006 *V* and 0.00015 *A*, respectively.

Figure 9 shows state estimation errors compared using SMO with gains optimized with robustness to disturbance parameter (H−). The state estimation error is sinusoidal variation with line frequency with the peak magnitudes of $V_{o_{d}q}$ and $I_{o_{d}q}$ being 8V and 0.14A, respectively.

Figure 10 shows state estimation errors compared using SMO with gains optimized with robustness to disturbance parameter (H−/H∞). The state estimation error is a sinusoidal variation with line frequency with the peak magnitudes of $V_{o_{d}q}$ and $I_{o_{d}q}$ being 0.1−0.4V and 0.025A, respectively.

Figure 11 shows the FTC performance for the d-component of sensor output current $\left(I_{d}\right)$ compared using SMO with gains optimized with fault sensitivity parameter (H−), robustness to disturbance parameter (H∞) and a mixed (H−/H∞) with non-faulty actual $I_{q}$. The results with (H−) are best among the other two, the mixed problem (H−/H∞) also stays relatively closer, whereas the results with (H∞) are more faulty.

Figure 12 shows FTC performance for q-component of sensor output current $\left(I_{q}\right)$ compared using SMO with gains optimized with fault sensitivity parameter (H−), robustness to disturbance parameter (H∞), and a mixed (H−/H∞) with non-faulty actual $I_{d}$. The results with (H−) are best among the other two, the mixed problem (H−/H∞) also stays relatively closer, whereas the results with (H∞) are more faulty.

Figure 13 shows the FTC performance for d-component of sensor output voltage $\left(V_{d}\right)$ compared using SMO with gains optimized with fault sensitivity parameter (H−), robustness to disturbance parameter (H∞) and a mixed (H−/H∞) with non-faulty actual $V_{q}$. The results with all three are comparable and non very differentiating w.r.t each other.

Figure 14 shows the FTC performance for the q-component of sensor output voltage $\left(V_{q}\right)$ compared using SMO with gains optimized with fault sensitivity parameter (H−), robustness to disturbance parameter (H∞), and a mixed (H−/H∞) with non-faulty actual $V_{q}$. The results with all three are comparable and not very differentiating w.r.t each other.

## 7. Conclusions

This paper has considered the VSI-based microgrid model as an application to apply the enhanced (in robustness or sensitivity to fault) sliding-mode observer for fault diagnosis and fault-tolerant control.The saturation faults of current/potential transformers (which are mounted in the passage of LCL filters are specifically considered, along with the general applicability of the approach for a good range of sensor/actuator faults;H∞ and H− parameters of robust control similar in nature to the game theoretic saddle points are used to derive the inequality version of HJIE, which is in terms of the Lyapunov function, faults, or disturbances, and the output error estimation vector as variables. The HJIE in inequality form not only proves the stability of observers but also gives the LMIs, which are convex optimized, to find the (H∞ and H−) constrained SMO gains providing the optimal/sub-optimal values of SMO gains ($G_{o}^{*}$, $G_{m}^{*}$, $f^{*}$, $\xi^{*}$), as mentioned in the pair of saddle points, i.e., ($Go^{*}$,$f^{*}$) and ($G^{*}$, $\xi^{*}$). The sliding-mode observer using the above gains in terms of the error vector is used to estimate the faults and disturbances;The main results computed are estimations of current/volatge faults of sensors (C.T/P.T), current/volatge fault estimation errors, and fault tolerance performance by the control block , which is provided by the corrections performed according to the faults estimated by SMOs, which are (i) robust to disturbance, (ii) sensitive to faults, and (iii) compromised of both.Moreover, all of the above mentioned results are given and compared for SMOs with the above three constraints. The gain optimization is accordingly done in Theorems 1–3, which is the main contribution of this research along with the applicability to composite faults (phase, magnitude, harmonics) occurring in sensors (C.T/P.T) mounted on LCL filters on the inverter outputs;The future works are intended to enhance the work for several microgrids operating in parallel, by using the applied fault tolerant control schemes in the distributed control paradigm, while managing the optimized power flow control between them. Moreover, the deep learning techniques can be opted as a future work in this domain.

## Figures and Tables

**Figure 1 sensors-22-01597-f001:**
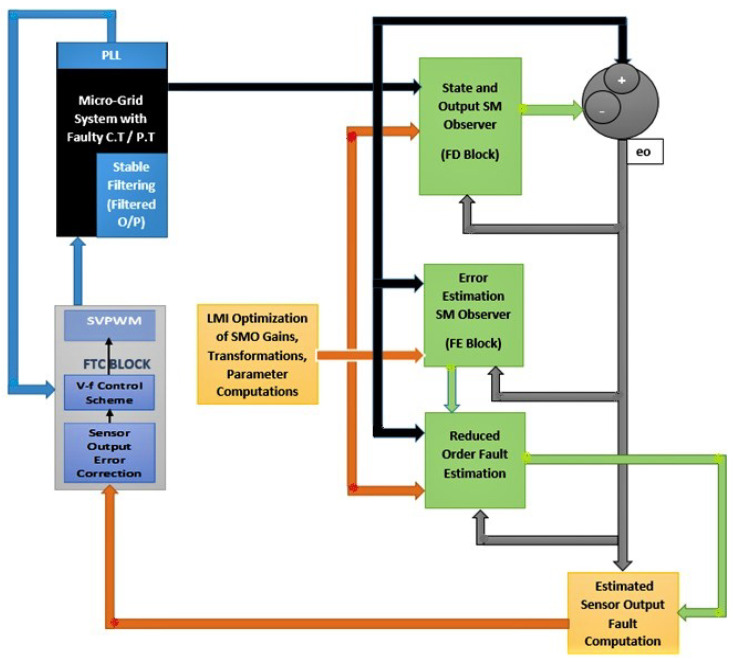
Complete block diagram of the used approach.

**Figure 2 sensors-22-01597-f002:**
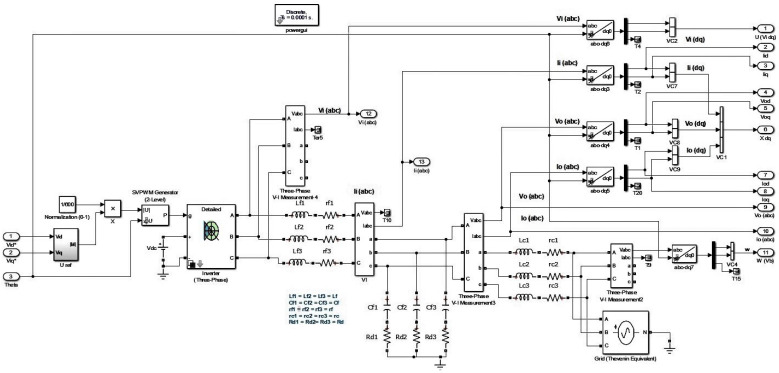
Wired microgrid model.

**Figure 3 sensors-22-01597-f003:**
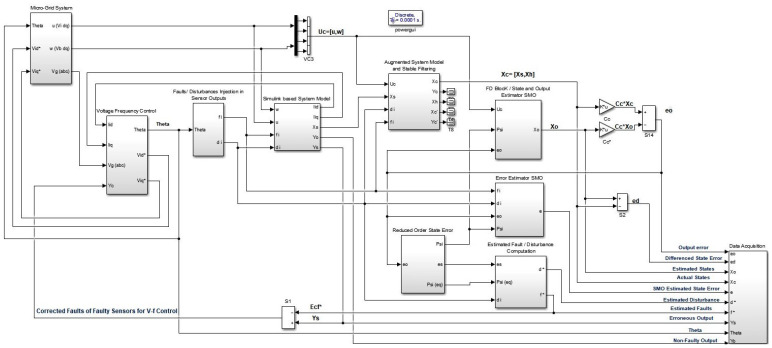
Simulink-based detailed block diagram of the used approach.

**Figure 4 sensors-22-01597-f004:**
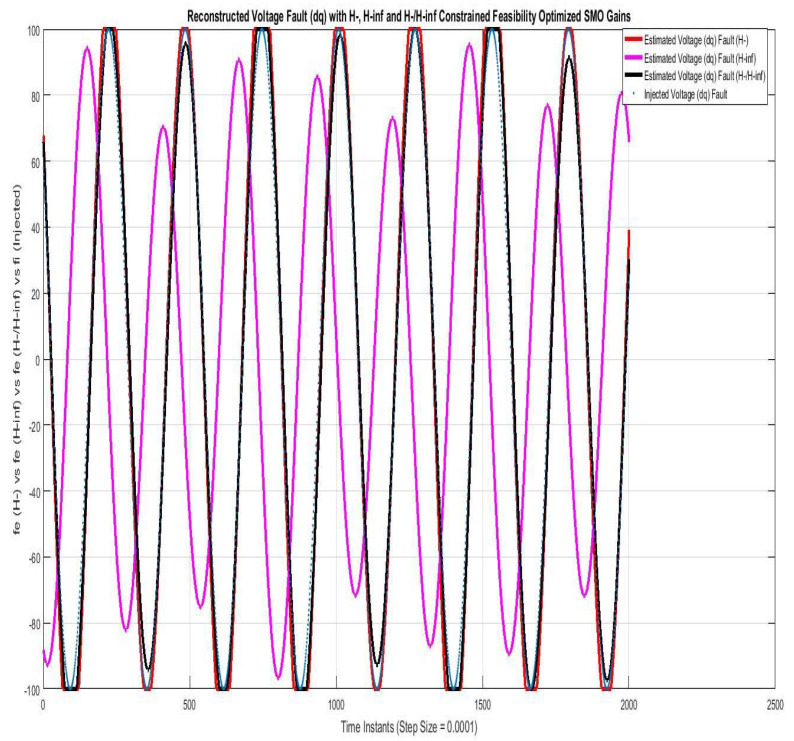
Reconstructed voltage fault (dq) with feasibility optimized (H−) and H−/H∞ SMO gains.

**Figure 5 sensors-22-01597-f005:**
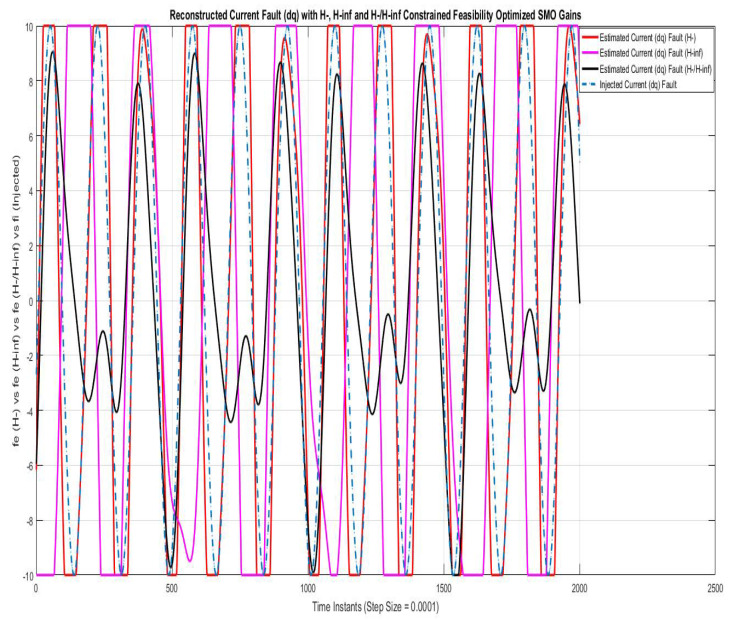
Reconstructed current fault (dq) with feasibility optimized H− and H−/H∞ SMO gains.

**Figure 6 sensors-22-01597-f006:**
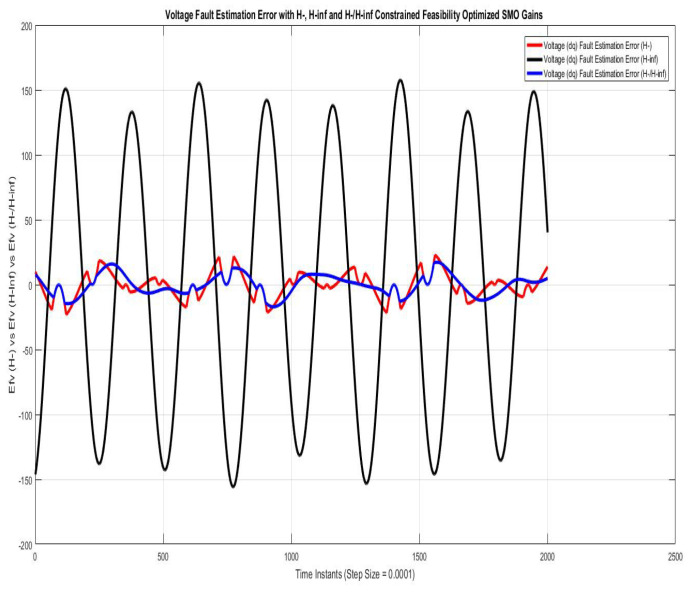
Voltage fault estimation error with feasibility optimized H− and H−/H∞ SMO gains.

**Figure 7 sensors-22-01597-f007:**
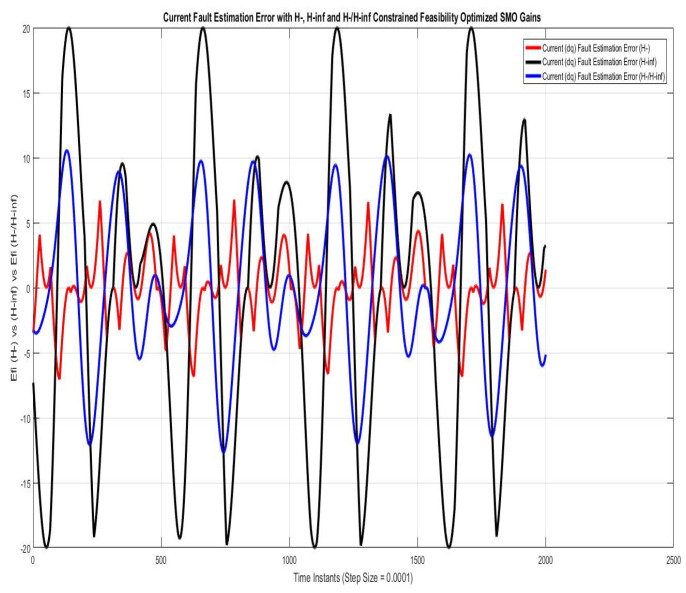
Current fault estimation error with feasibility optimized H− and H−/H∞ SMO gains.

**Figure 8 sensors-22-01597-f008:**
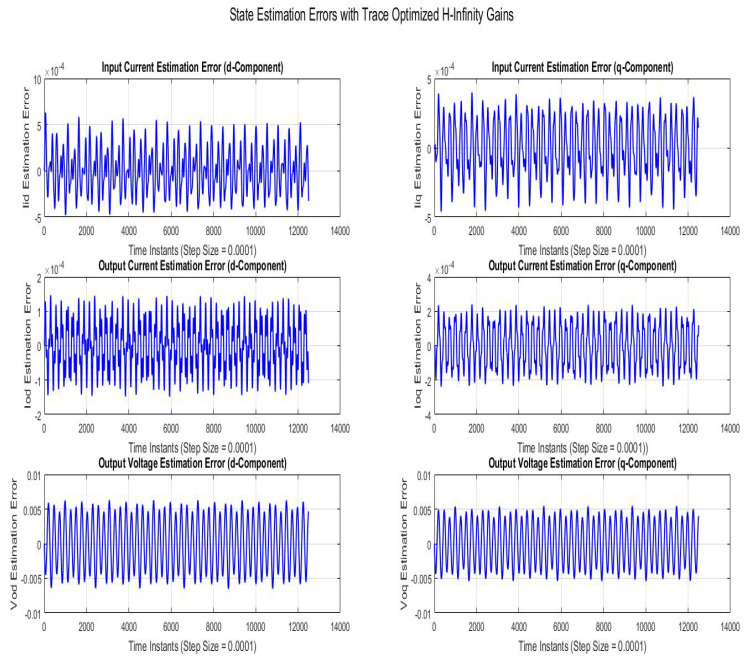
State estimation errors with trace optimized H∞ SMO gains.

**Figure 9 sensors-22-01597-f009:**
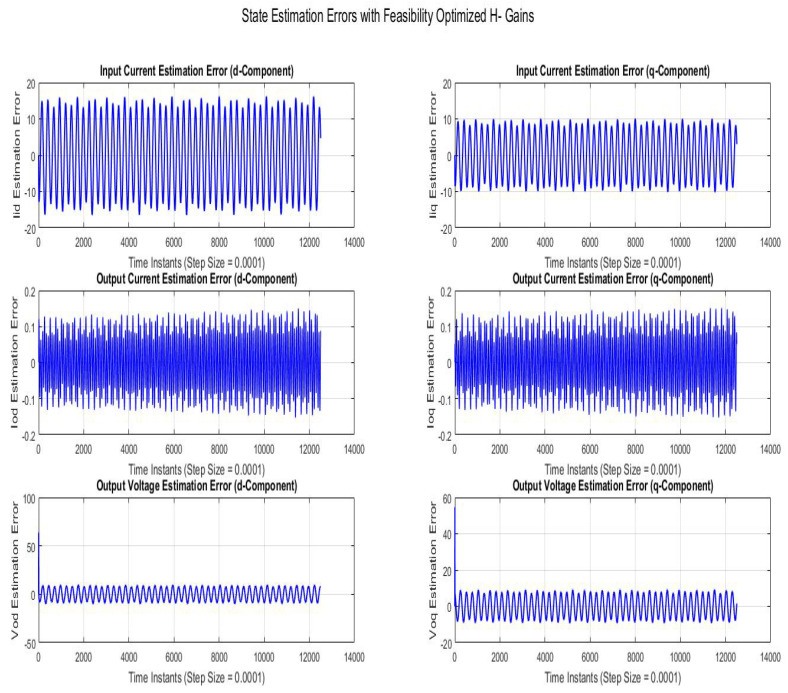
State estimation errors with feasibility optimized H− SMO gains.

**Figure 10 sensors-22-01597-f010:**
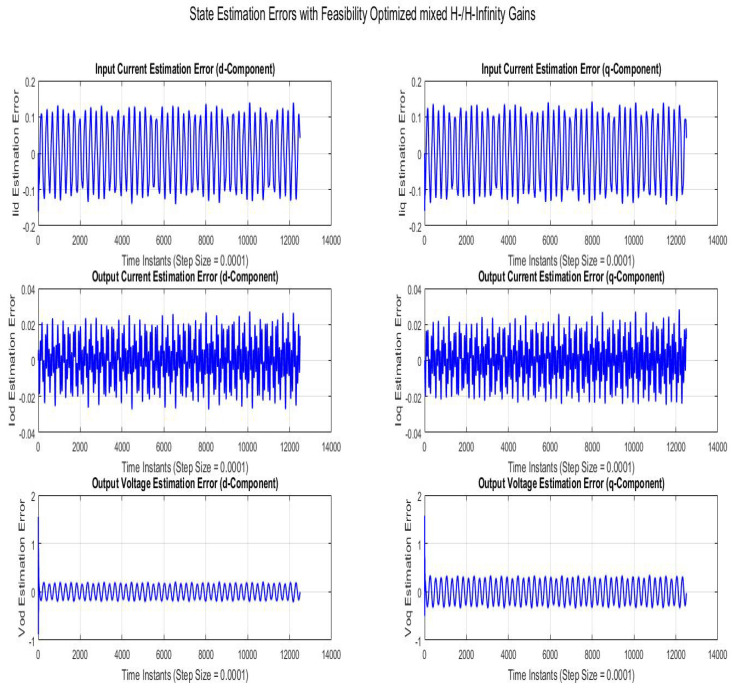
State estimation errors with feasibility optimized H−/H∞ gains.

**Figure 11 sensors-22-01597-f011:**
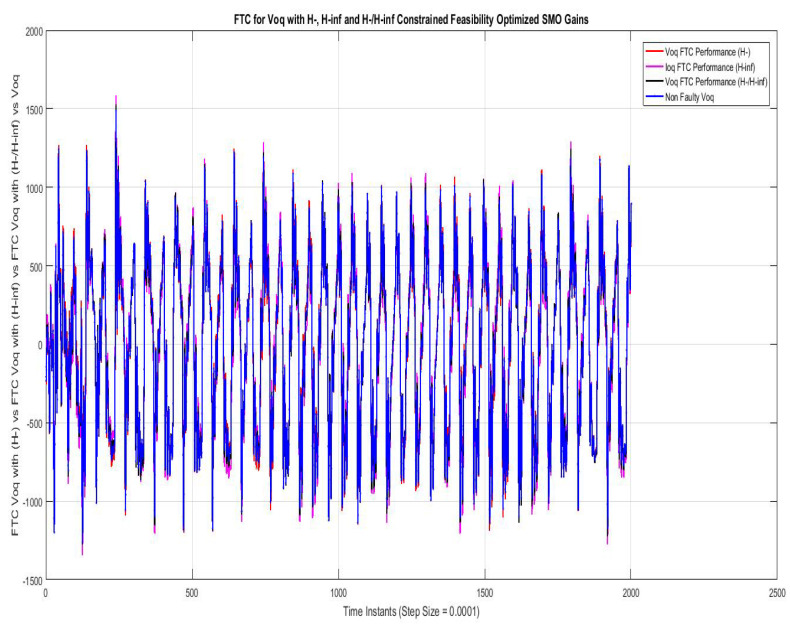
FTC for Ioq compared with feasibility optimized H− and H−/H∞ SMO gains.

**Figure 12 sensors-22-01597-f012:**
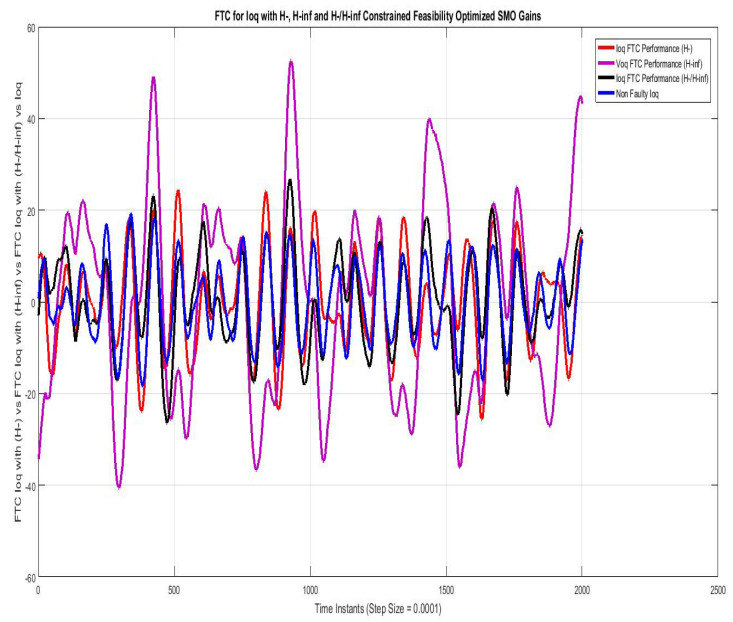
FTC for Ioq compared with feasibility optimized H− and H−/H∞ SMO gains.

**Figure 13 sensors-22-01597-f013:**
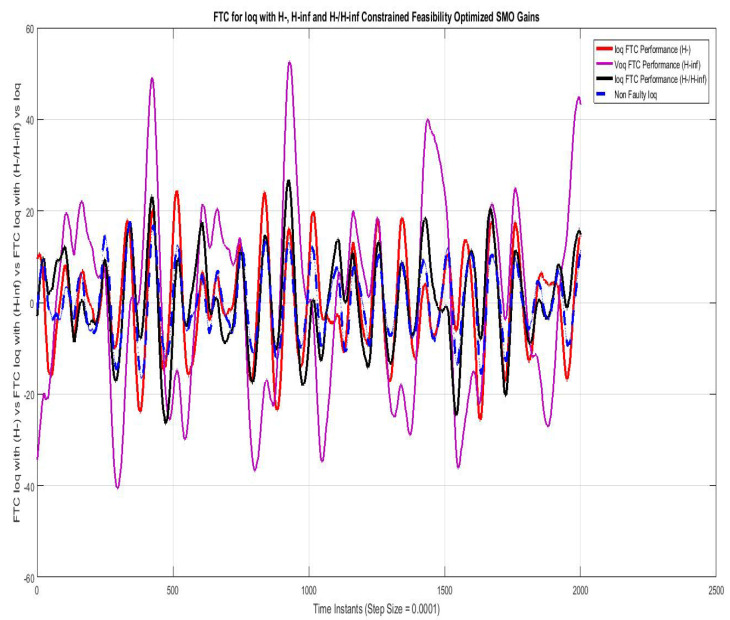
FTC for Vod compared with feasibility optimized H− and H−/H∞ SMO gains.

**Figure 14 sensors-22-01597-f014:**
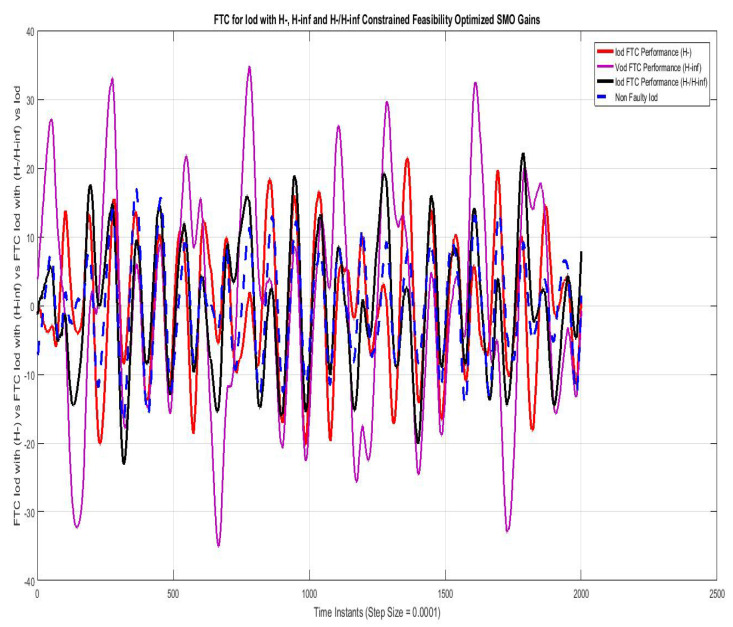
FTC for Voq compared with feasibility optimized H− and H−/H∞ SMO gains.

**Table 1 sensors-22-01597-t001:** Microgrid system parameters.

Parameter	Value	Parameter	Value	Parameter	Value
$V_{dc}$	600 V	$V_{g}$	600 V	$\theta_{grid}$	$60^{\circ }$
$L_{f1}$	4.20 mH	$L_{f2}$	4.20 mH	$L_{f3}$	4.20 mH
$L_{c1}$	0.50 mH	$L_{c2}$	0.50 mH	$L_{c3}$	0.50 mH
$C_{f1}$	15μF	$C_{f2}$	15μF	$C_{f3}$	15μF
$R_{d1}$	2.025Ω	$R_{d2}$	2.025Ω	$R_{d3}$	2.025Ω
$r_{f1}$	0.50Ω	$r_{f2}$	0.50Ω	$r_{f3}$	0.50Ω
$r_{f1}$	0.09Ω	$r_{f2}$	0.09Ω	$r_{f3}$	0.09Ω
$\omega_{c}$	50.26 rad/s	$\omega_{n}$	377 rad/s	$\omega_{PLL}$	377 rad/s
$V_{oqn}$	85 V	m,n	1/1000	$\omega_{c,PLL}$	7853.98 rad/s

## Data Availability

The data required for any detailed simulations, Matlab/Simulink-based optimization algorithms and simulations, and mathematical proofs will be provided, if the paper gets accepted or with the final version, in order to share the knowledge for general benefit.

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
