# Peer review of "Fault Diagnostics and Tolerance Analysis of a Microgrid System Using Hamilton–Jacobi–Isaacs Equation and Game Theoretic Estimations in Sliding Mode Observers"

_sensors, 2022, doi:10.3390/s22041597_

Round 1
Reviewer 1 Report
- There are some abbreviations that do not explain what it means in the text. I hope that you would check it.
- The variables can be categorized as vector variables and scalar variables. I hope that you would distinguish the variables. In general, vector variables are expressed with bold text, and scalar variables are italic text.
- There are some equations that make got confused. I hope that you would use mathematical symbols clearly.
- There are a lot of variables that make got confused. I hope that you would explain in the Nomenclature section categorized with subscripts.
Author Response
Response sheet has been added. thanks

Reviewer 2 Report
This article proposes an interesting approach for fault diagnosis and tolerance analysis of a microgrid. Using the Simulink model ensures that this method can be easily adapted to different electrical systems. However, I have some major concerns.
- The main objective of the paper should be expressed more clearly in the abstract and in the introduction.
- The introduction should be rearranged: the main objective of the paper should be clearly described; Table 1 should be deleted or moved to the end of the introduction section; the terms presented in Table 1 should be described when used in the text.
- Compared with the existing work, the contribution of this paper is not clear. It is recommended that the authors list the core contributions of this article in the introduction.
- The introduction should not be divided into sections: sections 1.1 and 1.2 could become two subsections of section 2.
- Sections 3 and 4 should be profoundly changed. The authors are advised to move most of the mathematical treatment not strictly related to the case study to one or more appendices at the end of the work.
- Section 4 must be made understandable by limiting the mathematical description (by inserting citations or by moving to the appendix) and highlighting the application to the case study.
- All figures 3 to 14 need to be graphically improved and moved to section 6.
- The conclusion section should summarize the procedure briefly and summarize the main results obtained. If possible, include possible future developments.
Author Response
Response sheet has been added. thanks

Reviewer 3 Report
This paper presents the use of game theory applied to robust control in order to detect faults in CT/PTs in a microgrid.
1- Please, do an English revision throughout the whole document.
2- At the title and at several parts of the document, HJIE is referring to Issacs. Is this correct? Some references in the internet refer to it as Isaacs. Please check.
3- I am not sure about this, but it seems to me that there is a maximum number of keywords. Please check author instructions.
4- Usually, MDPI journals have a special place to put nomenclature, at the end of the paper. I am not sure if the template has changed. Please check author instructions.
5- Although the acronyms are already defined in Table 1, I suppose they should be introduced in parenthesis the first time they appear in the main text. Please check author instructions and check the first occurrences of:
- CT/PT (line 27);
- LMI (line 34);
- LTV (line 39);
- MPC (line 47);
- FDI (line 48);
- SMO (line 57);
- ILMI (line 74);
- FAR and FDR (line 78);
- FTC (line 91);
6- line 42: isoability - Please check if this a typo.
7- There is a mistake in the phrase between lines 49 and 51 ("A paper in [7] uses sliding mode observer based technique is used for non-linear air craft system where uncertainty being function of state variables has a non-linear bound"). Perhaps the correct could be "A paper in [7] uses sliding mode observer based technique for non-linear air craft system where uncertainty being function of state variables has a non-linear bound" OR "A sliding mode observer based technique is used in [7] for non-linear air craft system where uncertainty being function of state variables has a non-linear bound".
8- line 49: Instead of "A paper in [7] uses...", why not use the name of the author? "Yan [7] uses..."
9- Please check if line 59 is correct: Hinfinity is repeated twice and the same for H-.
10- Please, consider the english revision of the two first phrases of subsection 1.1 .
11- What is the equation between lines 127 and 128? It should be numbered and introduced in the main text. For example, "Equation (1) presents..."
12- Please, introduce Figure 1 in the text. For example, "Figure 1 presents ...". Also, discuss the figure. What are its blocks?
13- line 185: missing a period punctuation mark.
14- The equation between lines 205-206 (Clarke/Park?): Please, number the equation and introduce it in the text.
15- At section 6, please, introduce Figure 3 in the text.
16- line 635: Table 2, instead of Table 1. Also, period punctuation mark is missing.
Author Response
Response sheet has been added. thanks

Round 2
Reviewer 1 Report
- The abbreviations and symbols are explained using Nomenclature. However, some symbols are not matched between Nomenclature and equations in terms of text types. In your manuscript, the symbols of vector variables in equations bold-italic. Therefore, I hope that you would check it.
- You showed the process of the approach using a block diagram. However, there is no description of the algorithm you used, HJIE and gam theoretical estimation. I hope that you would add the algorithms in the figure to be more detailed and easy to know.
- In Eq. (8), there is no explanation about A_h. I hope that you would check it.
Author Response
Response sheet attahced. thanks

Reviewer 2 Report
The new version of the paper is clearer and well written.
I have only one comment:
The conclusion section should be reorganized and the images 1-14 should be placed in other sections
Author Response
Response sheet attached. thanks
